# Laboratory measurements of stomatal $NO_2$ deposition to native California trees and the role of forests in the $NO_x$ cycle

Erin R. Delaria[1], Bryan K. Place[1], Amy X. Liu[1], and Ronald C. Cohen[1,2]

[1]Department of Chemistry, University of California Berkeley, Berkeley, CA, USA
[2]Department of Earth and Planetary Science, University of California Berkeley, Berkeley, CA, USA

**Correspondence:** Ronald C. Cohen (rccohen@berkeley.edu)

**Abstract.**

Both canopy-level field measurements and laboratory studies suggest that uptake of $NO_2$ through the leaf stomata of vegetation is a significant sink of atmospheric $NO_x$. However, the mechanisms of this foliar $NO_2$ uptake and their impact on $NO_x$ lifetimes remains incompletely understood. To understand the leaf-level processes affecting ecosystem scale atmosphere-biosphere $NO_x$ exchange, we have conducted laboratory experiments of branch-level $NO_2$ deposition fluxes to six coniferous and four broadleaf native California trees using a branch enclosure system with direct Laser Induced Fluorescence (LIF) detection of $NO_2$. We report $NO_2$ foliar deposition that demonstrates a large degree of inter-species variability, with maximum observed deposition velocities ranging from $0.15 - 0.51$ cm s$^{-1}$ during the daytime, as well as significant stomatal opening during the night. We also find that the contribution of mesophyllic processing to the overall deposition rate of $NO_2$ varies by tree species, but has an ultimately inconsequential impact on $NO_x$ budgets and lifetimes. Additionally, we find no evidence of any emission of $NO_2$ from leaves, suggesting an effective uni-directional exchange of $NO_x$ between the atmosphere and vegetation.

## 1   Introduction

Nitrogen oxides ($NO_x \equiv NO + NO_2$) are a form of reactive nitrogen that play a major role in the chemistry of the atmosphere. $NO_x$ catalyzes tropospheric ozone formation, contributes to the production of photochemical smog, and influences the oxidative capacity of the atmosphere (Crutzen, 1979). $NO_x$ is primarily emitted as NO through fossil fuel burning, lighting, and soil microbial activity (Seinfeld and Pandis, 2006). The latter source is of particular importance in remote forested, and agricultural regions, where emission from soils is the primary source of $NO_x$.(e.g. Jacob and Wofsy, 1990; Lerdau et al., 2000; Seinfeld and Pandis, 2006; Romer et al., 2018; Almaraz et al., 2018).

Understanding the fate of atmospheric $NO_x$, in addition to its emission sources, is essential for interpreting the impact of $NO_x$ on atmospheric chemistry. Prior studies have demonstrated that $NO_2$ can directly deposit to foliage after diffusion through stomata (e.g., Teklemmariam and Sparks, 2006; Chaparro-Suarez et al., 2011; Breuninger et al., 2013; Delaria et al., 2018). The currently understood mechanism of this uptake process is as follows: $NO_2$ enters through the stomatal cavity and dissolves into the apoplastic fluid, forming nitrate, which then is reduced to ammonium by the enzyme nitrate reductase (Park and Lee,

1988; Ammann et al., 1995; Tischner, 2000; Lillo, 2008; Heidari et al., 2011). There is evidence that $NO_2$ may also be directly scavenged by antioxidants, most notably ascorbate (Ramge et al., 1993; Teklemmariam and Sparks, 2006). These processes may be impacted by the leaf pH, which is known to change under conditions of limited water availability (Bahrun et al., 2002). Experiments using $^{15}N$ as an isotopic tracer have demonstrated that absorbed $NO_2$ is eventually assimilated into amino acids (Rogers et al., 1979; Okano and Totsuka, 1986). Although the role of stomatal conductance ($g_s$) in controlling the deposition

of $NO_2$ is well-documented, the impact of mesophyllic processes remains poorly resolved. These mesophyllic mechanisms are complex and include any process taking place between the intercellular air space and the ultimate nitrogen assimilation site. The question of whether and how much mesophyllic processes affect $NO_x$ budgets at the canopy scale thus persists.

The most divisive example of the mesophyll quandry is the sometimes-reported emission of $NO_x$ from plants, mostly in the form of NO, at low $NO_x$ mixing ratios that would be relevant to remote forested regions (Johansson, 1987; Rondón and

35 Granat, 1994; Hereid and Monson, 2001; Sparks et al., 2001; Teklemmariam and Sparks, 2006). This would, under many conditions, indicate that trees instead serve as a constant source, rather than sink, of $NO_x$. However, this idea has been called into question by a number of recent studies including Chaparro-Suarez et al. (2011), Breuninger et al. (2013) and Delaria et al. (2018). It is possible that the magnitude and direction of the $NO_x$ flux to leaves may vary depending on the species and conditions. One such factor that has been suggested to impact foliar emission and deposition of $NO_x$ is elevated soil

nitrogen. Soil nitrate fertilization has been documented to lead to an increase in nitrate reductase activity in the needles of scots pine seedlings (Andrews, 1986; Pietilainen and Lahdesmaki, 1988; Sarjala, 1991). It is possible that as a result of abundant nitrate fertilization, nitrate accumulates in leaves, leading to emission or a reduction in uptake. For example, Chen et al. (2012) observed an increase in NO emission and Teklemmariam and Sparks (2006) detected an increase of $NO_2$ emission under conditions of elevated soil nitrate. *Per contra*, Joensuu et al. (2014) found no evidence of fertilization-induced $NO_x$ emissions.

No influence of soil nitrogen on either $NO_2$ or NO uptake has been documented (Okano and Totsuka, 1986; Teklemmariam and Sparks, 2006; Joensuu et al., 2014).

In this study we present results from laboratory measurements of $NO_2$ fluxes of ten native California tree species–six conifers and four broadleaf trees–using a branch enclosure system and laser-induced fluorescence (LIF) detection of $NO_2$. Here we investigate the relative influence of stomatal and mesophyllic processes on the total uptake rate of $NO_2$ under atmospherically

relevant conditions. Our aim is to assess the factors controlling $NO_2$ foliar deposition and their ultimate impact on the $NO_x$ cycle. To test this, we measured the $NO_2$ deposition velocity over a range of stomatal conductances and considered evidence for additional limits on the uptake rate. We also conducted experiments under drought and elevated soil nitrogen and tested for indications of $NO_2$ emission or changes in the apparent mesophyllic uptake limit.

## 2 Methods

### 2.1 Tree specimens

Foliar deposition of $NO_2$ was investigated in the laboratory using ten native California tree species–*Pinus sabiniana, Pinus ponderosa, Pinus contorta, Pseudotsuga menziesii, Calocedrus decurrens, Sequoia sempervirens, Arbutus menziesii, Acer*

*macrophyllum, Quercus agrifolia, and Quercus douglasii.* Three to six individuals of each species were purchased from a local native California plant nursery (Native Here Nursery) or Forestfarm, where the plants were grown from seeds and cuttings.

The tree specimens were grown in a nutrient-rich commercial soil mixture of Sun Gro Sunshine #4 and Supersoil potting soil in 20—40 liter pots in an outdoor section of the Oxford facility greenhouse at the University of California, Berkeley. The trees were 2—3 years old when measurements were taken. No additional fertilizers or pesticides were used on the plants. Trees were transported into the lab for experimentation, where they were exposed to a 12 h light/dark cycle. Trees were illuminated with an LED diode array of 430—475 and 620—670 nm lights (Apollo Horticulture). For the deciduous trees (*Q. douglassi, and*

*A. macrophyllum*) experiments were run between May and September 2019. For all other species experiments were conducted year-round, between October 2018 and November 2019.

## 2.2 LIF measurement of NO$_2$ deposition fluxes

Measurements were made with a dynamic chamber and Laser-Induced Fluorescence (LIF) detection of NO$_2$. A full description of our apparatus can be found in Delaria et al. (2018). Briefly, an NO$_2$ standard was mixed with humidified zero air (air filtered

to remove NO$_x$ and reactive species) and delivered to a ~10 L chamber enclosing the branch of a tree at a total flow rate of ~6000 cm$^3$ min$^{-1}$ (Fig.1). The lifetime of air within the chamber was $\sim$ 2 min. Humidity was adjusted by controlling the fraction of zero air that passed through a bubbler filled with distilled water. The mixing ratios of NO$_2$ entering the chamber were typically between 0—10 ppb. Some of the air entering the chamber was diverted to cell #1 of the NO$_2$ LIF analyzer and two Licor instruments (6262 and 7000) for measuring the mixing ratios of NO$_2$ and H$_2$O/CO$_2$, respectively in the in-

flowing air stream, such that the flow rate of air directly into the chamber was ~5000 cm$^3$ min$^{-1}$. Air from the chamber was simultaneously pumped out to cell #2 of the NO$_2$ LIF analyzer and the Licor-7000 instrument for measuring the mixing ratio of NO$_2$ within the chamber and the change in CO$_2$ and water vapor between the in- and out-going air streams, respectively (Fig. 1). A slight positive pressure was maintained within the chamber to ensure lab air did not leak into the chamber.

Fluxes of NO$_2$ to leaves were calculated according to (Eq. 1—2):

$$Flux = \frac{Q}{A}([NO_2]_{in} - [NO_2]_{out}) \tag{1}$$

$$Flux = V_d([NO_2]_{out} - [NO_2]_{comp}) \tag{2}$$

where [NO$_2$]$_{in}$ and [NO$_2$]$_{out}$ are concentrations of NO$_2$ entering and exiting the chamber, respectively, at chamber equilibrium. Chamber equilibrium is achieved when the flow rates in and out of the chamber are equal and can be identified by a

85 constant concentration of [NO$_2$]$_{out}$. [NO$_2$]$_{comp}$ is the compensation point concentration, $Q$ is the flow rate (cm$^3$/s), $A$ is the enclosed one-sided leaf area, and $V_d$ is the deposition velocity. The leaf area was determined using the ImageJ software package (Schneider and Eliceiri, 2012) and the flow rate was measured at the beginning of each experimental run (Mesa Laboratories 510-M Bios Defender). Peroxyacetyl nitrate (PAN) and acetone were also delivered to the chamber for simultaneous measurements of PAN stomatal deposition. Negligible thermal production of NO$_2$ was observed. The results of PAN deposition

experiments will be discussed elsewhere. The $NO_2$ mixing ratio was also corrected for the differences in collisional quenching of the excited state $NO_2$ by water vapor in cells #1 and #2, caused by transpiration of the tree within the chamber (Thornton et al., 2000).

$$[NO_2]_{out,actual} = [NO_2]_{out,measured} \times (1 + 5\Delta X_{H_2O}) \tag{3}$$

where $\Delta X_{H_2O}$ is the difference in the water vapor mole fraction between the chamber and the incoming air stream. Experiments to an empty chamber were conducted approximately every two months during this study to calculate the deposition of $NO_2$ to the chamber walls. The wall loss was at maximum $\sim$2% of the $[NO_2]_{in}$ concentration and was background subtracted from our flux calculations.

Deposition velocities were determined using the method described in Delaria et al. (2018): a weighted orthogonal distance linear regression was performed on $NO_2$ fluxes (determined using Eq. 1) against $[NO_2]_{out}$ to obtain a slope equal to $V_d$. A positive x-intercept was interpreted as evidence for a possible compensation point. During each day of experimentation we stepped through at least 8 different $NO_2$ concentrations, with each concentration step lasting for 40 minutes. Uncertainty in $V_d$ was obtained through propagating uncertainty in measured $NO_2$ concentrations, $Q$, and $A$. The uncertainty in $NO_2$ concentrations was estimated as one standard deviation of variation in measurements during the last 10 minutes of each concentration step. The uncertainty in $Q$ was estimated as <1 % and a 10% uncertainty was estimated for the enclosed one-sided leaf area.

The deposition velocities measured can be related to the resistance-model framework for deposition of trace gases developed by Baldocchi et al. (1987) (Eq.4—6).

$$V_d = \frac{1}{R} \tag{4}$$

$$R = R_a + R_b + R_{leaf} \tag{5}$$

$$\frac{1}{R_{leaf}} = \frac{1}{R_{cut}} + \frac{1}{R_s + R_m} \tag{6}$$

$R$ is the total resistance to deposition, $R_a$ is the aerodynamic resistance, $R_b$ is the boundary layer resistance and $R_{leaf}$ is resistance to uptake by the leaf. $R_a$ was assumed to be negligible under our chamber conditions (Pape et al., 2009; Breuninger et al., 2012; Delaria et al., 2018). $R_{leaf}$ is made up of $R_{cut}$, $R_s$, and $R_m$. Respectively, these refer to the cuticular resistance (resistance to deposition to the surface of the leaf), stomatal resistance ($1/g_s$), and mesophyllic resistance (resistance associated with all processes taking place within the leaf that limit uptake).

## 2.3 Measurement of stomatal conductance

$CO_2$ and water vapor exchanges were measured using the Licor 6262 and Licor 7000 instruments. Measurements of water vapor exchange were used to calculate the transpiration rate ($E$) and total conductance to water vapor ($g_t^w$) using Eq. 7 and Eq.

8, according to von Caemmerer and Farquhar (1981).

$$E = \frac{Q}{A} \frac{w_a - w_e}{1 - w_a} \tag{7}$$

$$g_t^w = \frac{E(1 - (w_i + w_a)/2)}{w_i - w_a} \tag{8}$$

where $w_a$ and $w_e$ are the mole fractions of water vapor of the outgoing and incoming airstreams, respectively, and $\omega_i$ is the internal leaf water vapor mole fraction. $\omega_e$ was measured with the Licor-6262 with dry air as a reference and $\Delta\omega$ ($\omega_a - \omega_e$) was measured with the Licor-7000 with incoming air as the reference. $\omega_e$ was kept constant throughout a day of measurements and was varied between days. Measurements of an empty chamber were also used to calculate and correct for the water vapor deposition to the chamber at varying relative humidity. The difference between $\omega_a$ and $\omega_e$ for an empty chamber was not statistically significant and at all relative humidity levels was within instrumental uncertainty of the Licor-6262. $\omega_i$ was assumed to be the saturation vapor pressure at the leaf temperature, which was measured with a thermocouple at the surface of an enclosed leaf. The chamber temperature was measured with a second thermocouple and was typically $20\pm3°$C. Over the course of a day the temperature and humidiy varied by a maximum of 2 °C and 5%, respectively. These deviations were not found to be significantly correlated with stomatal opening. The photosynthetic photon flux density (PPFD) was monitored outside the chamber with a LiCor quantum sensor (LiCor LI-190SA) and was $1190\,\mu$mol m$^{-2}$ s$^{-1}$, approximately the PPFD for Berkeley, California, at noon during the month of October. We performed calculations based on von Caemmerer and Farquhar (1981) to confirm this is above the photon flux required to achieve maximal stomatal aperture for tree types relevant to this study. Total conductance was calculated as the average over the light or dark period of an experiment. The uncertainty in our calculation of total conductance to water vapor was primarily influenced by uncertainty in the leaf temperature and the assumption of leaf water vapor saturation. We observed fluctuations in the temperature of enclosed leaves of $\pm2°$C. Total uncertainty in $g_t^w$ was determined by propagating this uncertainty in leaf temperature, which resulted in larger estimated uncertainties at larger chamber humidities, usually coinciding with higher stomatal conductances. Chamber relative humidity was maintained at less than 90% to minimize this effect. Variations in stomatal conductance were achieved by varying the mole fraction of water vapor in the air delivered to the chamber. The Licor-6262 instrument was calibrated weekly using standard CO$_2$ cylinders and a Licor-610 dewpoint generator. The Licor-7000 instrument was calibrated daily.

The stomatal conductance ($g_s^w$) could then be calculated from Eq. 9:

$$\frac{1}{g_s^w} = \frac{1}{g_t^w} - \frac{1}{g_b^w} \tag{9}$$

where $1/g_b^w$ is the boundary layer resistance to water vapor. The boundary layer resistance to water vapor was estimated to be negligible under our experimental conditions, with an upper bound of 0.6 s cm$^{-1}$. This was calculated by measuring the deposition of NO$_2$ to a 30 cm$^2$ tray of activated charcoal and confirmed by measuring the evaporation from a water-soaked Whatman No. 1 filter paper. (Delaria et al., 2018). A detailed description of our assumption of negligible $R_b$ can be found in section 3.1. Stomatal ($g_s$) and total ($g_t$) conductances to NO$_2$ were calculated by scaling the values for water vapor by the ratio of diffusivities in air ($D_{NO_2}/D_{H_2O}$) according to Massman (1998).

## 2.4 Nitrogen measurements

To test the influence of excess soil nitrogen on the ability of trees to take up nitrogen through their stomata in the form of $NO_2$, we fertilized three individuals of both *Quercus agrifolia* and *Pseudotsuga menziesii* with a 20 mM ammonium nitrate solution. The trees were watered with 250 ml of this ammonium nitrate solution three days per week. Three individuals of each species were watered with DI water as the control group. The trees underwent this fertilization treatment for 120 days before beginning dynamic chamber measurements on $NO_2$ foliar deposition. $NO_2$ deposition experiments were conducted for 70 days, during which time the soil fertilization treatments were continued.

### 2.4.1 Soil nitrogen

Approximately 5 mg of a soil core sample was taken each day from the individual on which we conducted an $NO_2$ deposition experiment. The soil was sifted through a mesh 2 mm sieve. Soil nitrate and ammonium were extracted by shaking $\approx 2.5$ mg of the soil sample in 30 ml of $\approx 2$M KCl for one hour, followed by filtering the samples through a Whatman No.1 filter paper. The other $\approx 2.5$ mg was dried in a drying oven at 60°C for at least 48 hours. The mass of the soil after drying was measured to determine the percentage dry mass of the extracted soil sample. Six KCl blanks, 3 KCl samples spiked with 5 mL (low QC), and 3 KCl samples spiked with 10 mL KCl (high QC) were carried through the extraction process to serve as quality controls (QC samples). $NH_4^+$ and $NO_3^-$ were measured using a colorimetric synthesis following the method of Sims et al. (1995) and Decina et al. (2017). Briefly, a standard 1 ppm stock solution of ammonium nitrate was made from ammonium nitrate solid dissolved in milli-q water, and was diluted to 0, 0.1, 0.2, 0.3, 0.4, and 0.5 mg/L in 1 cm, 2.5 mL cuvettes. These standard solutions served as the calibration standards; we made three sets of calibration standards for both ammonium and nitrate analysis. All glassware was acid washed in a 1M solution of HCl prior to all measurements and extractions to prevent contamination.

For ammonium analysis, 160 $\mu$L of each soil extraction sample from the control group, 10 $\mu$L from the fertilizer-treated group, and 1.6 mL of the QC samples were pipetted into individual cuvettes. 100 $\mu$L of 0.2 M citrate , 200 $\mu$L of 5 mM nitroprusside, 100 $\mu$L of 0.3 M hypochlorite reagents, and 500 $\mu$L of milli-q water were then added sequentially into each cuvette. The cuvettes were filled to a final volume of 2.5 mL with KCl, and the samples were allowed to sit for 30 min. For nitrate measurements, 320 $\mu$L and 10 $\mu$L of soil samples from the the control and fertilized groups, respectively, and 1550 $\mu$L of the QC samples, were pipetted into separate cuvetts. 950 $\mu$L of a regent containing 1g/L vanadium chloride and 25 mg/L N-(1-Naphthyl)ethylenediamine (NEDD) was subsequently added to each cuvette, which were then filled to a final volume of 2.5 mL with KCl and allowed to sit for 24 hrs. 160 $\mu$L and 320 $\mu$L of a control *Q. agrifolia* soil extraction sample were added to one set of calibration standards for ammonium and nitrate analysis, respectively, to test the effects of the soil matrix on the calibration.

Concentrations of ammonium and nitrate in each sample were determined with colorimetric measurements using a custom built spectrophotometer. The spectrophotometer light source was a broad spectrum quartz tungsten-halogen lamp (QTH10 Thorlabs Inc.). The absorption of each sample and standard was measured with the light source passing through a $540 \pm 2$

nm bandpass filter (FB570-10 Thorlabs Inc.) for nitrate analysis or a $670 \pm 2$ nm bandpass filter (FB540-10 Thorlabs Inc.) for ammonium analysis.

### 2.4.2 Uncertainty analysis

Concentrations of ammonium and nitrate in the soil extraction samples were determined from the slope in their respective calibration curves. The calibrations for ammonium and nitrate analysis had respective uncertainties of 7% and 5%. The slopes of the calibration curves with added sample from a *Q. agrifolia* soil extraction were not statistically different from those containing only standards, allowing us to exclude the possibility of interference from the soil matrix.

The accuracy uncertainty in the high and low QC samples were 3% and 11%, respectively for anmmonium measurements, and 3% and 12% for nitrate measurements. We estimated the resulting uncertainty for cuvette samples with less than 0.15 mg/L $NH_4^+$ or $NO_3^-$ ($\approx 1.8$ $\mu$g/mg soil $NH_4^+$ or $NO_3^-$) to be 15%. Samples with larger concentrations were estimated to have 5% uncertainty. The blank quality control standards contained 0.04 mg/L ammonium and nitrate. This was blank-subtracted from each sample.

### 2.4.3 Leaf nitrogen

After deposition experiments were completed the leaves were removed from the trees and dried for 48 hours in a drying oven. The leaves were then ground to a fine powder and the percent nitrogen, hydrogen, and carbon content were measured with a ICP Optima 7000 DV instrument.

### 2.5 Drought stress

*Calocedrus decurrens* and *Pinus ponderosa* were drought stressed to study the impact of drought on $NO_2$ deposition. Three individuals of each species were watered daily (control group) and three individuals of each species were watered with 250 mL once every four weeks (drought group). Limited-water treatment of the drought group was carried out for 60 days before conducting dynamic chamber experiments for $NO_2$ foliar deposition. $NO_2$ deposition experiments were run for 30-40 days. During the experiments, the control group was watered 50 mL daily and the experimental group was watered 50 mL once every two weeks. The *P. ponderosa* drought-stress experiments took place between March and June 2019. The *C. decurrens* drought stress spanned from August to December 2019.

The xylem water potential ($\Psi_p$) of the trees were monitored to measure the drought stress level of the trees using a Scholander pressure chamber (Model 670 PMS Instr. Comp.). Leaves were cut, wrapped in aluminum foil, and then inserted into the pressure bomb. The $\Psi_p$ of cuttings were measured around 11:00 AM each day. A $\Psi_p$ measurement lower than -1.0 MPa indicated signs of drought stress in the *P. ponderosa*. The *C. decurrens* did not show evidence of drought stress in $\Psi_p$ measurements while in the greenhouse, however, early signs of embolism were observed.

## 3 Results

$V_d$ was calculated for each day of measurements with a weighted linear regression of measured fluxes and chamber $NO_2$ concentrations (Delaria et al., 2018). No statistically significant compensation point was observed under any experimental condition for the majority of the species studied, in agreement with previous work (Chaparro-Suarez et al., 2011; Breuninger et al., 2013; Delaria et al., 2018). Only *P. menziesii* was found to have a compensation point, estimated to be 20 ppt, but this concentration is below the limit of quantification for our instrument so we consider this measurement to be consistent with a

compensation point of zero. $V_d$ and $g_s$ measurements allowed for consideration of whether the deposition of $NO_2$ is exclusively stomatally controlled, or is also affected by the internal processing in the mesophyll. We rarely observed total closing of the stomata when the chamber lights were turned off at night. All of the deposition observed at night could be explained by deposition to these partially open stomata. This is consistant with previous studies observing only partial closing of stomata at night in a variety of plant species (Dawson et al., 2007; Drake et al., 2013) . The results of experiments are shown in (Table 2).

### 3.1    Measurements of mesophyllic resistance

We utilized two methods for analysing the importance of the mesophyllic resistance on the deposition of $NO_2$. Figure 2 shows the predicted stomatal-limited $NO_2$ deposition fluxes, assuming negligible $R_b$, $R_c$, and $R_m$ ($Flux = g_t[NO_2]_{out}$) plotted vs. the measured $NO_2$ fluxes. Our upper bound measurement of $R_b$ for $NO_2$ was 1 s cm$^{-1}$ (0.6 s cm$^{-1}$ for water vapor). Assuming $g_s = g_t$ would lead to a maximum of a 60% or 10% error in the calculated $g_s$ with a $g_t$ = 0.6 cm s$^{-1}$ or $g_t$ = 0.1

230    cm s$^{-1}$, respectively. However, $R_b$ decreases with the enclosed leaf area according to Pape et al. (2009), which at a minimum was 200 cm$^2$. The maximum $R_b$ in the chamber should have thus been $\approx$0.1 s cm$^{-1}$. Assuming $g_s = g_t$ would lead to a maximum of a 6% error at $g_t$ = 0.6 cm s$^{-1}$ in this case. Any deviation from unity in the observed slope of predicted vs. measured fluxes can thus be attributed to $R_m$. Any error in our assumption of negligible $R_b$ may partially mask the effect of $R_m$. We do not expect that variation in $R_b$ due to changes in leaf morphology, micrometeorology, and leaf movement would

substantially change the effect of $R_b$, although we cannot rule out the possibility that this was partially responsible for day-to-day fluctuations in $NO_2$ fluxes. We confirmed the validity of our assumption of negligible $R_b$ by comparing measurements of total conductance to water vapor, $g_t^w$, in the chamber to measurements of stomatal conductance for the enclosed branch with a Licor-6800 instrument under identical environmental conditions of light irradiation, humidity, and temperature. This test was performed on one individual of three different tree species, and in all cases the chamber $g_t^w$ measurements were found to be

approximately equal to the Licor-6800 measurements of $g_s^w$ within the range of uncertainty in $g_t^w$.

Significant deviations from unity in the slope of $g_t[NO_2]_{out}$ vs measured fluxes could be seen in several species, most notably *S. sempervirens* (Table 2 and Fig. 2). Figure 2 shows each flux measurement as a single data point. For each day of experiments a slope of predicted vs. measured fluxes was obtained from a least squares cubic weighted fit on the 8—12 fluxes measured at varying $NO_2$ concentrations. The reported slope for a given species (Table 2, shown in blue in Fig. 2)

was calculated using a weighted average of the slopes from all experiment days. This was done to minimize the contribution of systematic errors potentially introduced by the Licor 7000 instrument, which was calibrated daily. All data points for a

given day were excluded (shown in red in Fig. 2) if the calculated slope on that day was determined to be an outlier by a generalized extreme studentized deviate test for outliers. Identified outliers were excluded both to account for potentially erroneous deviations in the $V_d/g_t$ ratio (most likely due to systematic error in calibration of the Licor-7000 instrument), and to avoid over-weighting of days with abnormally large stomatal conductances. These latter instances normally coincided with low $V_d/g_t$ ratios, and if these data were also subject to some systematic error, would bias our analysis of $R_m$.

$R_m$ was also explicitly calculated using the relationship of $V_d$ and $g_t$. Figure 3 shows $V_d$ from each day of experiments plotted against the measured $g_t$. Positive y-intercepts are indications of cuticular deposition and curvatures in the fit away from the 1:1 line are implications of mesophyllic resistance. $R_m$ was calculated with a weighted fit of the resistance model:

$$V_d = \frac{1}{R_c} + \frac{1}{(\frac{1}{g_s} + R_m)} \tag{10}$$

No evidence of cuticular deposition was observed so only results of $R_m$ are recorded (Table 2). The deposition observed with the chamber lights turned off could be explained completely by the measured stomatal conductance. Fits of the resistance model (Eq. 10) typically resulted in cuticular resistances on the order of 1000 s cm$^{-1}$. $R_m$ was calculated both assuming negligible $R_b$ ($g_s = g_t$) and $R_b = 1$ s cm$^{-1}$. There were no significant differences between these two calculations (Table 2).

## 3.2 Effects of excess soil nitrogen

The impact of soil fertilization on the foliar uptake of $NO_2$ by two tree species, *Q. agrifolia* and *P. menziesii*, was examined by watering a control group of both species with deionized water and a fertilized group with 20 ppm ammonium nitrate. On average, the soil nitrogen concentrations of $NH_4^+$ and $NO_3^-$ were 100x larger for the fertilized groups than the control groups (Table 1). The percentage of leaf nitrogen content approximately doubled between the control groups and the fertilized groups (Table 1).

The effect of soil nitrogen fertilization and leaf nitrogen content on the ratio of $V_d/g_t$ is shown in Fig. 4. No significant relationship ($\alpha = 0.01$) was observed for either *Q. agrifolia* or *P. menziesii*, suggesting the mesophyllic processing of $NO_2$ is unaffected by soil or leaf nitrogen content . We also observe no increase in the compensation point of $NO_2$ as a result of higher leaf nitrogen content or elevated soil nitrogen (Fig. 5).

## 3.3 Drought stress measurements

The impact of drought stress on $NO_2$ foliar uptake for *C. decurrens* and *P. ponderosa* was observed by regularly watering a control group and watering an experimental, drought group at much lower frequency (once every 4 weeks in the greenhouse, and once every 2 weeks in lab). The median $\Psi_p$ measured was lower for the drought groups than the control groups (Table 3). *C. decurrens* drought median $\Psi_p$ was -0.80 MPa compared to control median of -0.30 MPa, and *P. ponderosa* drought median was -1.05 MPa compared to control median of -0.60 MPa. The first quartiles of the control groups and third quartiles of the drought groups did not overlap, reflecting a significant difference between the $\Psi_p$ measurements of the two groups. We also observed a strong correlation between measured $\Psi_p$ and stomatal conductance. We found a more substantial impact of drought on the water potentials, and of the water potentials on the stomatal conductance, in *P. ponderosa* trees than *C. decurrens*. Both

these California conifer species are quite drought resistant (Pharis, 1966; Kolb and Robberecht, 1996; Maherali and DeLucia, 2000), but these results may indicate *C. decurrens* is particularly protected against water loss.

The mesophyllic resistance ($R_m$) calculated showed a statistically significant difference for both *C. decurrens* and *P. ponderosa* between drought-stressed and control groups. $R_m$ in drought-stressed *C. decurrens* increased from 0.37 s cm$^{-1}$ to 1.17 s cm$^{-1}$, while in *P. ponderosa* $R_m$ decreased from 0.86 s cm$^{-1}$ to 0 s cm$^{-1}$ (Fig. S5).

## 4 Discussion

### 4.1 Effects of mesophyll resistance on the lifetime of NO$_x$

The mesophyllic resistances ($R_m$) for each of the ten tree species measured are calculated from Fig. 3 and Eq. 10 and are tabulated in Table 2, assuming either $g_s = g_t$ or the upper bound for $R_b$. The slopes of predicted fluxes vs. measured fluxes, calculated in Fig. 2, are also tabulated in Table 2. The importance of the mesophyllic resistance and internal processing of NO$_2$ can be evaluated by examining both $R_m$ and the slope of measured vs. predicted fluxes. We also examined the potential impact of the mesophyllic processing of NO$_2$ by considering the Pearson's correlation coefficient between $g_t$ and the slope for an individual experiment (1 day of light or dark data) of measured vs. predicted fluxes (Fig. S3). These correlation coefficients can be found in Table 2. The more negative this correlation, the greater the deviation in the slope from unity for higher values of $g_t$, consistent with larger impact of the mesophyll on the NO$_2$ uptake rate. All tree species except for *C. decurrens*, *Q. agrifolia*, and *Q. douglasii* show statistically significant correlations ($\alpha = 0.05$) (Table 2). $R_m$ becomes more important at larger stomatal conductances (lower stomatal resistances), as can be seen with the increasing deviations from 1:1 in some species at higher values of $g_t$ in Fig.3. Thus, even for trees with higher calculated $R_m$, the impact of mesophyllic processing is unlikely to be large if the maximum stomatal conductance observed is relatively small, resulting in a slope in the measured vs predicted flux that does not deviate greatly from unity. This is the case for *Q. agrifolia* and *P. ponderosa*. Alternatively, *P. sabiniana* demonstrates a case of a relatively small $R_m$, but also a smaller slope in measured vs. predicted fluxes, driven by consistently larger stomatal conductances (lower $R_s$) (Fig. 3). The most sizable impacts of mesophyllic NO$_2$ processing ares seen in *S. sempervirens*, *P. sabiniana*, and *A. macrophyllum*. These species have the largest maximum observed $g_t$ (Fig. 3, Talbe 2) and slopes of measured vs. predicted fluxes of $0.79 \pm 0.04$, $0.84 \pm 0.03$ and $0.84 \pm 0.03$, respectively. However, the greater uncertainty in measurements of stomatal conductance at a larger chamber humidity calls in to question the accuracy of many $g_t$ measurements larger than approximately 0.4 cm s$^{-1}$.

To evaluate with greater certainty the relationship of $V_d$ and $g_t$, we conducted a set of experiments in helium to raise the stomatal conductance by increasing the gas diffusivities while maintaining relatively lower chamber humidity. These experiments were conducted on four of the tree species: *P. sabiniana*, *S. sempervirens*, *Q. agrifolia*, and *A. menziesii*. In these experiments the $V_d/g_t$ ratio for *P. sabiniana* remained close to 1:1 up to 1.3 cm s$^{-1}$ stomatal conductance (Fig. 3). Experiments in helium for this species thus suggest a smaller contribution of the mesophyll (red dashed line in Fig. 3). $R_m$ calculated including helium experiments was not statistically different for *S. sempervirens, Q. agrifolia*, nor *A. menziesii*.

Our laboratory-measurements of mesophyllic resistance address the uncertainty in the literature for whether reactions in the mesophyll may be consequential for $NO_2$ deposition velocities. To our knowledge, no previous studies have explicitly calculated the mesophyllic resistance. Differences between leaf-level deposition velocities and stomatal conductances measured by Breuninger et al. (2013), and observations of leaf ascorbate impacts on uptake rates by Teklemmariam and Sparks (2006) have indicated mesophyllic reactions may be important. Additional studies (Gut et al., 2002; Eller and Sparks, 2006; Chaparro-Suarez et al., 2011) have also shown some evidence that between 20% and 40% of $NO_2$ deposition is under mesophyllic control. Our findings, however, suggest nearly 90% of uptake is controlled by the stomata.

Currently, atmospheric models incorporate a mesophyllic resistance to $NO_2$ of 0.1 s cm$^{-1}$ (Zhang et al., 2002). This would result in slope of measured vs. predicted fluxes of 0.94, even with a relatively large average $g_t$ of 0.6 cm s$^{-1}$. The median slope measured in our study was 0.89. Using the multibox canopy model presented in Delaria and Cohen (2020), we investigated whether our results could possibly imply a more important impact of the mesophyllic resistance on the atmospheric fate of $NO_x$ at the canopy level. This model takes into account in-canopy processes (e.g. vertical transport, chemistry, etc.) to scale leaf-level processes to the canopy-level. The model was run using meteorological conditions for June measured during the BEARPEX-2009 campaign, located at a ponderosa pine forest in the western foothills of the Sierra Nevada mountain range (38°58'42.9"N, 120°57'57.9"W, elevation 1315 m). The model was initialized over two days and data from the third day was analyzed. We conducted two model runs at a stomatal conductance ($g_s$) to $NO_2$ deposition of 0.3 cm s$^{-1}$–the median measured maximum stomatal conductance excluding *P.sabiniana*–with an $R_m$ of either 0.1 or 0.6 s cm$^{-1}$–the median measured $R_m$ excluding *P. sabiniana*. For a stomatal conductance to $NO_2$ of 0.3 cm s$^{-1}$ ($\approx$ 0.5 cm s$^{-1}$ to water vapor) the model predicts only a 2.5% decrease in $NO_x$ lost to deposition with an $R_m$ of 0.6 compared with an $R_m$ of 0.1 s cm$^{-1}$. The lifetime to deposition with an $R_m$ of 0.1 and 0.6 s cm$^{-1}$ was 30.5 hr and 32.2 hr, respectively, representing only a 6% difference. The total atmospheric lifetime of $NO_x$ in the boundary layer with an $R_m$ of 0.1 and 0.6 s cm$^{-1}$ was 4.86 hr and 4.89 hr, respectively, representing only a 0.6% difference. Even the observed seemingly substantial mesophyllic resistance of *S. sempervirens* is therefore likely to be irrelevant at the canopy-scale. Contributions from mesophyllic processing, though mechanistically important at a cellular level, are likely to not matter at the canopy-scale in California forests. We therefore suggest that on canopy and regional scales, mesophyllic processes within leaves of trees represent a negligible contribution to $NO_x$ budgets and lifetimes in California. More studies on crops, grasses, and North American tree species from outside of California are needed.

### 4.2 Effects of excess soil nitrogen

We observed no effects of soil nitrogen, in the form of $NH_4^+$ and $NO_3^-$, or the leaf nitrogen content on the ratio of $V_d/g_t$ (Fig. 4) for either *Q. agrifolia* or *P. menziesii*. Changes in this ratio would indicate an effect on the mesophyllic resistance. We did observe declines in $g_t$ in the fertilized group relative to the control group during the later stages of experimentation, which coincided with observable evidence of plant stress (e.g., browning, wilting, and beginning signs of embolism). All variation in the uptake rates ($V_d$) could be explained exclusively with deviations in $g_t$. These results are supported by previous studies which have also found a negligible impact of nitrogen fertilization on $NO_2$ uptake (Teklemmariam and Sparks, 2006; Joensuu et al., 2014). If the fertilizer results in increased $NO_3^-$ and $NO_2^-$ in the leaf, this suggests that the mechanism of $NO_2$ uptake via

dissolution and subsequent reduction of $NO_3^-$ and $NO_2^-$ is likely not reversible and not influenced by accumulation of $NO_3^-$ and $NO_2^-$ within the mesophyll. Alternatively, if the increase in soil nitrogen leads only to an accumulation of organic nitrogen in the leaf, this increase has no effect on the uptake rates. Numerous studies indicate nitrate reductase activity is affected by the presence of ammonium, nitrate and organic nitrogen in the form of amino acids in a variety of plant species (e.g. Datta et al., 1981; McCarty and Bremner, 1992; Woodin et al., 2006). Based on our current understanding of the mechanism of $NO_2$ mesophyllic processing, if reactions in the mesophyll indeed affect the rate of stomatal uptake, our fertilization experiments should have succeeded in changing $NO_2$ uptake rates, given that they succeeded in changing leaf nitrogen content. Because we observed no effect of nitrogen fertilization on $NO_2$ uptake, we believe that this finding further supports that reactions within the mesophyll may be atmospherically unimportant. It is also possible that the disproportionation of $NO_2$ to form nitrate and nitrite and scavenging by antioxidants (e.g. ascorbate) are the rate limiting steps in the mesophyllic processing of $NO_2$, rather than enzyme activity. More leaf and cellular-level studies are needed to elucidate the uptake mechanism.

We also did not observe any evidence for a relationship between the $NO_2$ compensation point and the soil nitrogen content nor the leaf nitrogen content (Fig 5) for either *Q. agrifolia* or *P. menziesii*. In general, we only observed uptake and no emission of $NO_2$. We also conducted measurements of NO uptake and emission, but the fluxes measured were so small they were below the limit of quantifcation for our instrument. Chen et al. (2012) observed a strong relationship between NO emissions from stomata and soil nitrate fertilization. However, the maximum NO emissions they measured were a factor of 50 lower than the deposition of $NO_2$ measured here. NO emission from leaves is therefore not likely to be an important source of atmospheric $NO_x$. *P. menziessi* was the only tree examined in our experiments that demonstrated any evidence for emission of $NO_2$ at low mixing ratios, with a compensation point of $\approx 20$ ppt. This concentration is much lower than has been observed in previous studies that have detected an $NO_2$ compensation point (Hereid and Monson, 2001; Teklemmariam and Sparks, 2006). However, this concentration is near the limit of detection for our instrument (Delaria et al., 2018) so should be taken *cum grano salis*. A possible cause for discrepancy between our study and those that have measured significant $NO_2$ compensation points is that our experiments are conducted only using photosynthetically active radiation. Some past work has demonstrated that UV light may cause photolysis of nitrate at the leaf surface and subsequent emission of $NO_x$ (Hari et al., 2003; Raivonen et al., 2006). The lack of a relationship between $NO_x$ emission and soil N fertilization contrasts with the results of Teklemmariam and Sparks (2006), but is consistent with the nitrogen fertilization experiments conducted by Joensuu et al. (2014).

### 4.3 Effects of drought stress

Although there was a statistically significant impact of drought stress on $R_m$, this is unlikely to be important to the overall uptake rates of $NO_2$ an the canopy scale for reasons discussed in section 4.1. The differing effects of drought on $R_m$ between *P. ponderosa* and *C. decurrens* is surprising, with the drought group having a smaller $R_m$ in *P. ponderosa* and larger $R_m$ in *C. decurrens*. However, in the case of *P. ponderosa*, the lack of measurements at larger $g_t$ is likely to mask any existing mesophyllic effects, leading to minimal deviation in the total slope of predicted vs. measured fluxes from unity (Fig. S5). Despite a calculation of significant mesophyllic resistance in both drought and control *C. decurrens* individuals, the lack of a statistically significant ($\alpha = 0.05$) correlation between $g_t$ and the slopes of predicted vs. measured fluxes casts doubt on this

relationship. The control group of *P. ponderosa* is the only for which this correlation is significant. The impact of drought on NO$_2$ uptake at the leaf-level is thus primarily its effect on the stomatal conductance. At the canopy-level, documented effects of drought on leaf area also requires consideration (Pharis, 1966; Kolb and Robberecht, 1996; Maherali and DeLucia, 2000).

### 4.4 Effects of nighttime stomatal deposition

Most atmospheric chemical transport models, such as the abundantly utilized WRF-Chem and GEOS-Chem, use the Wesley model for parameterizing dry deposition of gaseous species (e.g., Skamarock and Powers, 2008; Fast et al., 2014; Amnuaylo-jaroen et al., 2014; Ng et al., 2017). The Wesley model implicitly assumes the stomata are fully closed at night, despite more recent studies demonstrating many species of vegetation maintain partially open stomata at night (Musselman and Minnick, 2000; Dawson et al., 2007; Fisher et al., 2007; Drake et al., 2013). We find minimal cuticular deposition of NO$_2$, in agreement with several other studies (Sparks et al., 2001; Chaparro-Suarez et al., 2011). However, field observations have shown that substantial leaf-level nighttime deposition of NO$_2$ is necessary to explain nighttime levels of NO$_x$ (Jacob and Wofsy, 1990). The same phenomenon has been seen with other gaseous molecules, most notably PAN, which has also been suggested by a number of field observations to have significant non-stomatal deposition at night (Turnipseed et al., 2006; Wolfe et al., 2009; Crowley et al., 2018). Sparks et al. (2003) did not observe any evidence of non-stomatal deposition in the laboratory, but more recently Sun et al. (2016), implicated non-stomatal deposition in accounting for over 20% of PAN leaf-level deposition. Our PAN deposition experiments however, discussed in Place et al. ES&T in press, also did not identify any significant non-stomatal deposition. Despite the existing differences regarding the importance of non-stomatal PAN deposition, we suggest that a significant portion of the "missing" deposition sink of NO$_2$ and peroxyacyl nitrates at night may be due to non-total closure of the stomata.

To assess the impact of nighttime stomatal opening on the atmospheric fates and lifetimes of NO$_x$ at night, we ran our 1-D multibox canopy model, under the conditions described above, at the minimum, maximum, 25th percentile, and 75th percentile of the median nighttime deposition velocities measured in this study (0.004, 0.087, 0.009, and 0.038 cm s $^{-1}$, respectively). At such low stomatal conductances, we found these deposition velocities to be not significantly different ($\alpha = 0.05$) from the stomatal conductance to NO$_2$. The fractions of NO$_x$ loss to deposition and chemistry to these levels of stomatal opening at night are shown in Fig. 6. Here chemistry represents loss to HNO$_3$, RONO$_2$, and PAN, and nighttime is defined from 20:00 — 05:00. The range between the first and third quartile of the nighttime deposition observed results in a range in the fraction of NO$_x$ loss to deposition from 13% to 25% (Fig 6) and a range in total NO$_x$ lifetime from $\approx$ 7.5—5 hrs.

The relatively large impact of the nighttime stomatal conductance on the fate of NO$_x$, coupled with the large degree of inter-species variation in nighttime stomatal opening, indicates a need for more extensive studies of the nighttime deposition of NO$_2$. Deposition is a permanent sink of atmospheric NO$_x$, contrasting with chemical nighttime sink of NO$_x$ to peroxyacyl nitrates (Russell et al., 1986; Cantrell et al., 1986; Perring et al., 2009). Heterogenous reactions at aerosol surfaces involving the NO$_x$ reservoir N$_2$O$_5$ and alkyl nitrate formation are among the other major nighttime chemical NO$_x$ sinks (Perring et al., 2009; Stavrakou et al., 2013; Kenagy et al., 2018). The relative fractions of nighttime NO$_x$ loss to deposition and chemistry is likely to have a substantial impact on the fate of atmospheric NO$_x$ and the cycling of reactive nitrogen.

## 4.5 Impacts on the nitrogen cycle in California

To our knowledge, this is the first study conducted on $NO_2$ stomatal deposition to native California tree species, except for *Q. agrifolia* (Delaria et al., 2018). However, there are many measurements of the stomatal conductance of California trees (Table 4) with which to compare our maximum total conductance to water vapor measurements (max $g_t^w$ ). Murray et al. (2019) examined patterns in maximum stomatal conductance to water vapor (max $g_s^w$ ) across bioclimatic zones. Among the species they looked at were *A. menziesii*, *A. macrophyllum* and *Q. agrifolia*, for which they measured an average max $g_s^w$ of 550 mmol $m^{-2}$ $s^{-1}$, 420 mmol $m^{-2}$ $s^{-1}$, and 390 mmol $m^{-2}$ $s^{-1}$, respectively. In comparison, our measurements of max $g_t^w$ for these species were, respectively, $210 \pm 10$ mmol $m^{-2}$ $s^{-1}$, $400 \pm 100$ mmol $m^{-2}$ $s^{-1}$, and $90 \pm 20$ mmol $m^{-2}$ $s^{-1}$. Our estimates of max $g_t^w$ for *A. menziesii* and *Q. agrifolia* are substantially lower. Matzner et al. (2003) report larger conductances than we do for *Q. douglasii* as well (Table 4). Maire et al. (2015) determined a maximum stomatal conductance for *A. menziesii* of 150 mmol $m^{-2}$ $s^{-1}$, in better agreement with our measurements. Henry et al. (2019) measured a similar maximum stomatal conductance of *Q. agrifolia* to our study of 95 mmol $m^{-2}$ $s^{-1}$, also in better agreement with our results than Murray et al. (2019). Maire et al. (2015) measured a maximum stomatal conductance to water vapor for *P. ponderosa* and *S. sempervirens* of 124 mmol $m^{-2}$ $s^{-1}$ and ~91 mmol $m^{-2}$ $s^{-1}$, respectively–considerably smaller than the values measured in this study. Ambrose et al. (2010) measured a max $g_s^w$ for *S. sempervirens* of 240 $m^{-2}$ $s^{-1}$, in better agreement with our measurements. *C. decurrens* max $g_t^w$ reported here are in good agreement with previous measurements of max $g_s^w$(Grantz et al., 2019). For *Quercus* and *Acer* species in similar climate regions to California, Maire et al. (2015) calculated max $g_s^w$ ranging from 103—890 mmol $m^{-2}$ $s^{-1}$ and 112—320 mmol $m^{-2}$ $s^{-1}$, respectively. The median of max $g_t^w$ for all four angiosperms we measured was 200 mmol $m^{-2}$ $s^{-1}$ , in good agreement with the 250 mmol $m^{-2}$ $s^{-1}$ median of all angiosperms in Mediterranean climate regions found by Murray et al. (2019) and the 215 $m^{-2}$ $s^{-1}$ median found by Maire et al. (2015). Our median for the six gymnosperms measured was 230 $m^{-2}$ $s^{-1}$, considerably larger than the median 100 $m^{-2}$ $s^{-1}$ max $g_s^w$ found by Maire et al. (2015) in Meditteranean climate regions (defined as warm temperature steppe regions as classified by Kottek et al. (2006)).

Overall, the total conductances to water vapor measured in our laboratory experiments fall within the ranges of maximum stomatal conductances measured in previous studies–although inconsistencies exist in the current literature. (We also consider this to further support our conclusion that the boundary layer resistance in our chamber is negligible). Possible discrepancies may have resulted from the location each species were measured, growing conditions, ages of the trees, etc. Nevertheless, our $NO_2$ deposition results–and their applicability to California forests–are bolstered by the fact that our max $g_t^w$ measurements fall within the ranges of max $g_s^w$ measured for for mature trees in the field. To assess the impact of the lab-measured deposition velocities on the $NO_x$ cycle in California, we used our measurements of maximum $V_d$ during the day and median $V_d$ at night ( $V_d^{max}$ and $V_d^{med}$(night), respectively) to estimate the flux and lifetime of $NO_x$ to deposition in forests throughout the state during the day and night, respectively (Fig. 7, Fig.8 ).

The average deposition flux to trees in California was calculated via Eq.11

$$F_{dep} = [NO_2] \times V_d^{eff} \times LAI \tag{11}$$

Leaf area index (LAI) data for June 2018 was obtained from MCD15A2H Version 6 Moderate Resolution Imaging Spectro-radiometer (MODIS) Level 4 product (Myneni et al., 2015) (Fig. S6). The $NO_2$ surface concentrations and planetary boundary layer heights over California were obtained from a WRF-CHEM simulation for June 2014 (Fig. S6) (Laughner et al., 2019). The month of June was chosen because in California this is when forests have a large LAI, large GPP, the greatest sunlight availability, and ecosystems often experience water limitations in the later summer (Turner et al., 2020). Land cover data was obtained from NLCD Land Cover (CONUS) for 2016 (Yang et al., 2018) (Fig. S1). Only forested sites were considered. Although the use of products from different years may introduce some error into our calculations, this will not qualitatively change our conclusion. Tree counts were obtained from the USDA Forest Service Forestry Inventory Analysis Database (for, 2014) (Fig. S2). For each approximately 24 $km^2$ hexagonal plot (Bechtold, 2005) in the Forest Service Inventory that contained more than 50% of the trees measured in our study, an effective deposition velocity to $NO_2$ ($V_d^{eff}$) was calculated as a weighted (by tree species abundance) average from the $V_d^{max}$ values listed in Table 2 (Fig. S6). Plots that contained less than 50% of the trees measured were not considered. Data was interpolated to a 500m grid. The resulting midday fluxes throughout California are shown in Fig. 7 and midnight fluxes are shown in Fig. 8. The greatest fluxes predicted are south of the San Francisco Bay Area, where there are high $NO_x$ concentrations, and also a relatively high forest LAI for an urban region (Fig. S6). Similar hotspots can be seen near Los Angeles in the inland chaparral regions. Large fluxes are also predicted in the foothill forest region of the Sierra Nevada mountain range, where there is a a large LAI, and frequent occurances of *P. sabiniana*, the tree having the largest $V_d$ (Fig. S2, Fig. S6). Relatively large fluxes occur in this region particularly during the nighttime.

The resulting lifetime of $NO_2$ to deposition was calculated via Eq. 12

$$\tau_{dep} = PBL \left( V_d^{eff} \times LAI \right)^{-1} \tag{12}$$

where PBL is the planetary boundary layer height. The lifetimes to deposition during the day are shown in Fig. 7. In forested regions the lifetime to deposition is approximately 10 hrs. This relatively short lifetime may be especially consequential in south of San Francisco Bay, where deposition could be competitive with the chemical sinks of $HNO_3$ and $RONO_2$ formation, which typically represent a lifetime to $NO_x$ loss of 2-11 hrs (e.g., Nunnermacker et al., 2000; Dillon et al., 2002; Alvarado et al., 2010; Valin et al., 2013; Romer et al., 2016; Laughner and Cohen, 2019).

The deposition fluxes and lifetimes to deposition during the night are shown in Fig. 8. With reduced deposition velocities at night, the nighttime deposition flux and the resulting total loss of $NO_2$ to deposition is small. However, with a reduced boundary layer during the night, the lifetime of $NO_x$ to deposition at night is on the same order as the deposition lifetime during the day (10—100 hr) and the overall $NO_x$ lifetime at night. This indicates this loss pathway may be an important nighttime sink of $NO_x$ from the atmosphere and may affect the nighttime chemical $NO_x$ sinks of alkyl nitrate formation and $N_2O_5$ chemistry (Brown et al., 2004, 2006; Crowley et al., 2010).

The estimations provided here are intended only to suggest qualitative indications of where $NO_x$ deposition may be important. Because we are ignoring effects of vertical transport and light attenuation through the canopy, and because we are using maximum measured deposition velocities, the deposition reported here is likely to be an upper-bound estimate. We recommend

areas where this estimated deposition is highest as regions that should be the subject of future field and large-scale modelling studies.

## 5 Conclusions

We present measurements assessing the relative effects of stomatal diffusion and mesophyllic processing of $NO_2$ on the uptake rate of $NO_2$. We find that the deposition velocity of $NO_2$ is essentially equal to the stomatal conductance to $NO_2$ under conditions of drought, excess soil nitrogen, variations in relative humidity, and in both the day and night. We find no evidence of any emission of $NO_2$ from leaves. $NO_2$ foliar exchange is thus uni-directional and variations are driven–from an atmospheric perspective–nearly entirely by the rate of diffusion through open stomata. This opens the possibility of using direct measurements of stomatal conductance–coupled with models and measurements of chemical transport, known relationships of the effects of environmental conditions on stomatal opening, measurements of canopy conductance, as well as indirect measurements–such as satellite solar-induced fluorescence data–to infer $NO_x$ foliar exchange. Additionally, we find significant differences in deposition velocities between species, reflecting differences in maximum stomatal conductance measurements that have been found by a number of previous studies (e.g., Ambrose et al., 2010; Maire et al., 2015; Henry et al., 2019; Murray et al., 2019). This diversity is not reflected in current atmospheric models, and may have a meaningful impact on estimates of regional $NO_x$ fluxes and lifetimes. Our observations of stomatal opening in the absence of light also suggest foliar deposition may represent as much as 25% of the total $NO_x$ loss at night, with stomatal deposition velocities as high as 0.038 cm s$^{-1}$. These findings not only have important implications for $NO_x$ chemistry, but are also relevant for the atmosphere-biosphere exchange of other gasses, such as $CO_2$ and biogenic volatile organic compounds.

*Author contributions.* ERD and BKP designed the experimental setup and ERD, BKP, and AXL collected all $NO_2$ exchange data. BKP and ERD designed methods and collected data for nitrogen fertilization experiments. ERD and AXL designed methods and collected data for drought stress experiments. ERD performed data analysis, with assistance from AXL. ERD prepared the manuscript in consultation with RCC. RCC supervised the project.

*Competing interests.* The authors declare that they have no conflict of interest.

*Data availability.* The data collected in this study can be obtained from the authors upon request.

*Acknowledgements.* We would like to thank Dr. Stephen Decina for his assistance in designing methods for soil ammonium and nitrate measurements. We would also like to acknowledge Dr. Robert Skelton for consultation on drought stressing trees and for allowing us to borrow a pressure chamber instrument for use in this study.

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

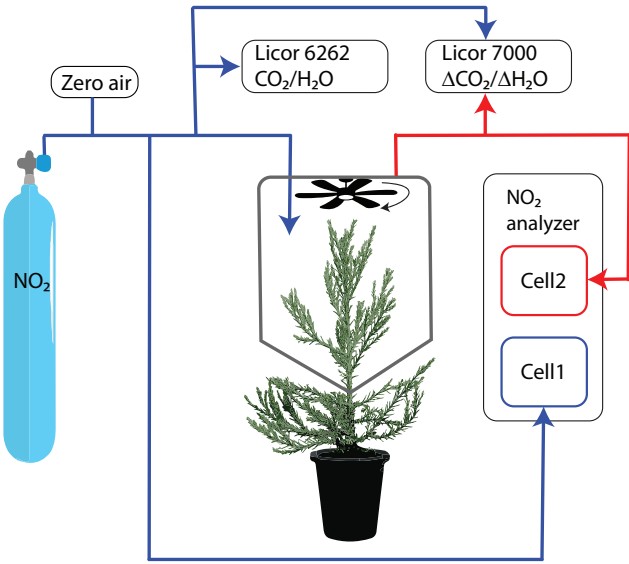

**Figure 1.** Figure of instrumental setup. Blue lines show the flow of gas that enters the chamber and red lines show the flow of gas sampled from the chamber.

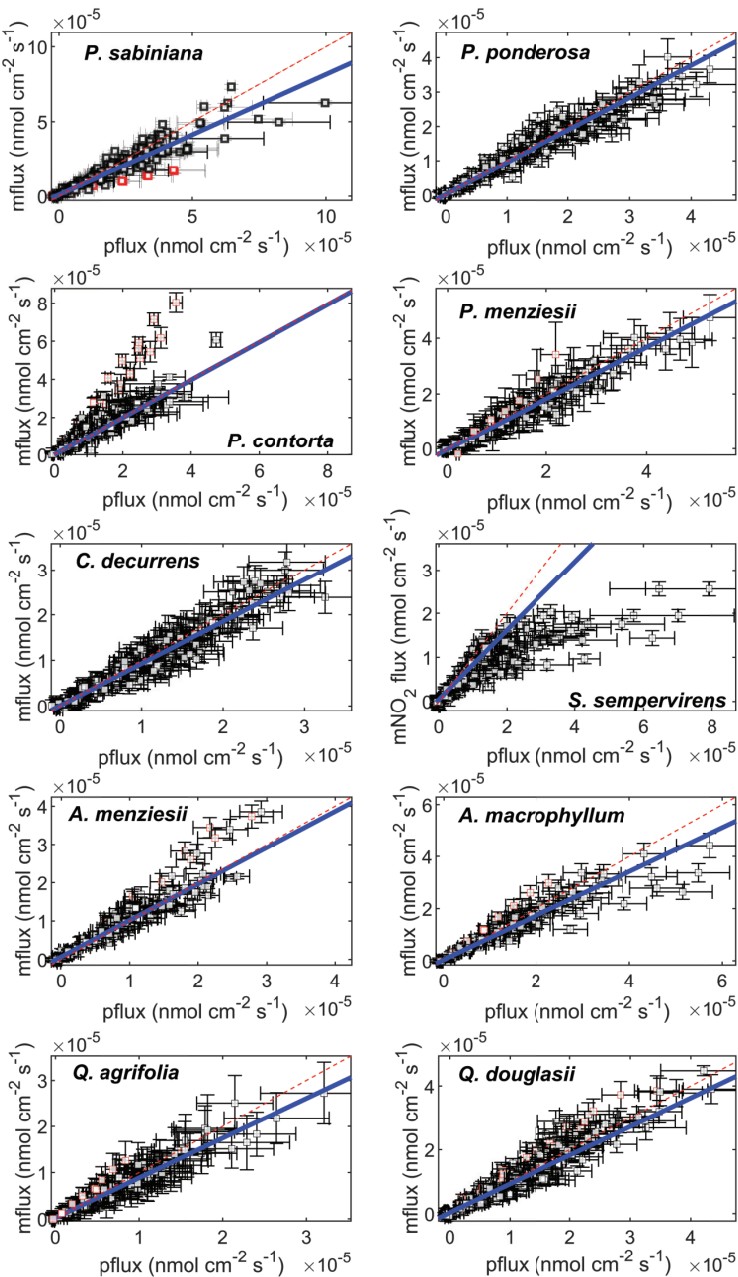

**Figure 2.** Measured fluxes (mflux) plotted against stomatal-limited predicted fluxes (pflux = $g_t[NO_2]_{out}$). Drought data and nitrogen fertilization data are included. Blue solid lines are the linear fit to data. Red lines are the 1:1 line. Error bars for the measured fluxes are calculated by propagating uncertainty in the measured $NO_2$ mixing ratios, the flow rate, and the leaf area (Eq. 1). Error bars for the predicted fluxes are calculated by propagating uncertainties in the measured $NO_2$ mixing ratios and the total conductance (Eq. 8). Red markers indicate data determined to be outliers by a generalized extreme studentized deviate test for outliers.

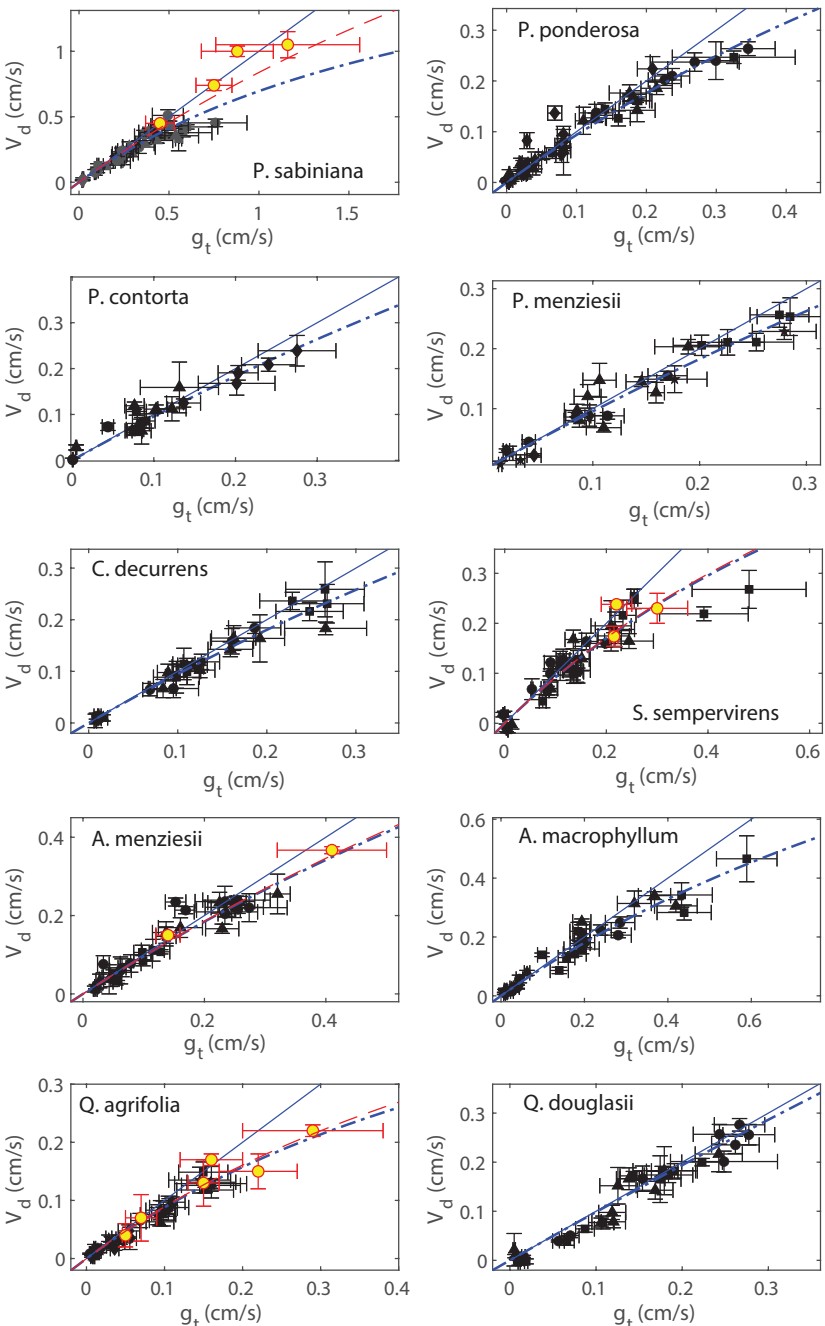

**Figure 3.** Deposition velocities ($V_d$) plotted against measured total conductances to $NO_2$ ($g_t$). Black markers represent measurements in zero air and red-yellow markers are measurements in helium. Measurements in helium are subject to less uncertainty introduced by potential systematic error in the leaf temperature. Solid blue lines are the 1:1 line and dashed blue lines are error weighted fits to the resistance model using only measurements in zero air, assuming the boundary layer resistance is negligible (Eq. 4). Fits to the resistance model including data from helium measurements are shown as dashed red lines.

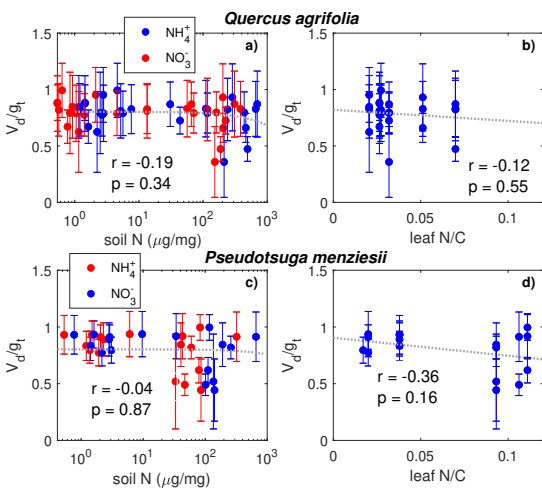

**Figure 4.** The $V_d/g_t$ ratio is plotted against soil nitrogen concentration in the form of $NH_4^+$ and $NO_3^-$ for (a) *Q. agrifolia* and (c) *P. menziesii*. The dashed line shows a linear fit to $NH_4$ data. The relationship is not significantly different ($\alpha = 0.05$) when fit to $NO_3^-$ data. The $V_d/g_t$ ratio is plotted against the leaf nitrogen:carbon ratio for (b) *Q. agrifolia* and (d) *P. menziesii*. $V_d/g_t$ ratios less that 1 imply contributions from the mesophyll to the $NO_2$ uptake rate. On each pannel the Pearson's correlation coefficient and the p-value for the slope are shown. The amount of soil and leaf nitrogen has no significant impact on the $V_d/g_t$ ratio.

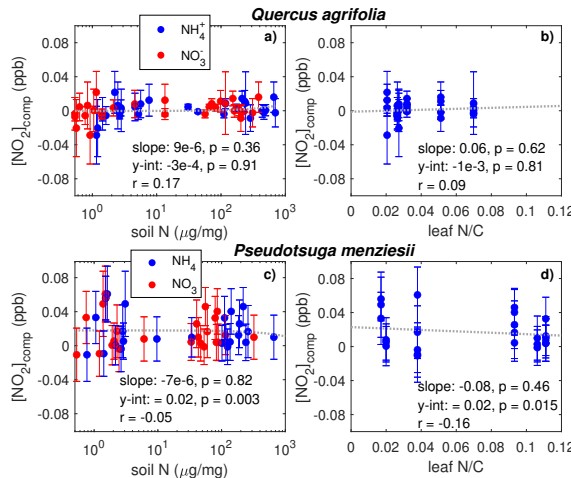

**Figure 5.** The concentration below which leaves emit $NO_2$ is the compensation point ($[NO_2]_{comp}$). $[NO_2]_{comp}$ is plotted against the soil nitrogen concentration in the form of $NH_4^+$ and $NO_3^-$ for (a) *Q. agrifolia* and (c) *P. menziesii*. The dashed line shows a linear fit to $NH_4$ data. The relationship is not significantly different ($\alpha = 0.05$) when fit to $NO_3^-$ data. $[NO_2]_{comp}$ is plotted against the leaf nitrogen:carbon ratio for (b) *Q. agrifolia* and (d) *P. menziesii*. On each panel the Pearson's correlation coefficient, the slope, the intercept, and their p-values are shown. The amount of soil and leaf nitrogen has no significant impact on the compensation point.

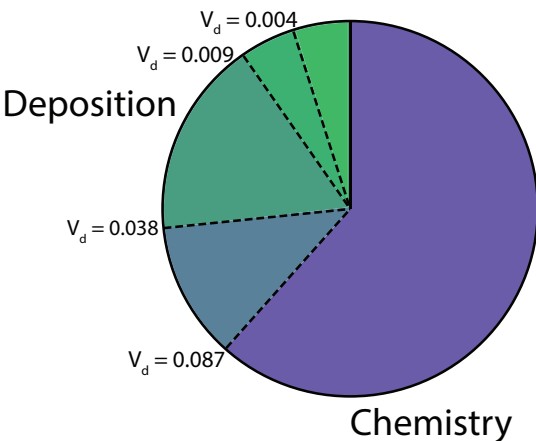

**Figure 6.** Fraction of NO$_x$ loss to deposition and chemistry (nitric acid, alkyl nitrate, and peroxyacyl nitrate) at night (20:00—05:00). The four dashed lines between the deposition and chemistry fractions show NO$_x$ loss with a nighttime NO$_2$ deposition velocity of 0.004, 0.009, 0.038, and 0.087 cm s$^{-1}$. These deposition velocities respectively represent the minimum, first quartile, third quartile, and maximum of the median nighttime deposition velocities measured for the native California trees examined in this study.

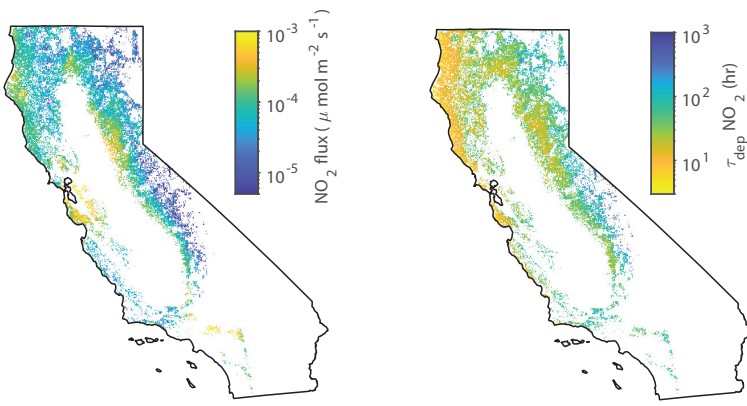

**Figure 7.** (left) Average midday deposition fluxes of NO₂ to forests in June throughout California. (right) Average midday deposition lifetimes of NO$_x$ in June throughout California. White areas are non-forested areas.

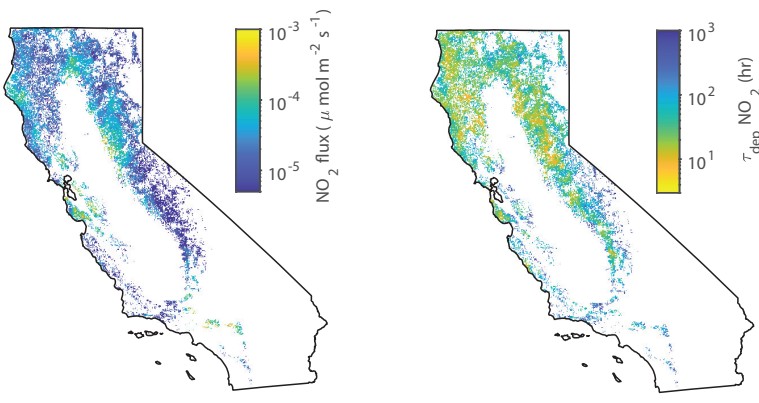

**Figure 8.** (left) Average midnight deposition fluxes of $NO_2$ to forests in June throughout California. (right) Average midnight deposition lifetimes of $NO_x$ in June throughout California. White areas are non-forested areas.

**Table 1.** Average soil and leaf nitrogen

| tree[a] | soil $NH_4^+$ | soil $NO_3^-$ | leaf N | leaf C |
|---|---|---|---|---|
| | $\mu$g/mg | $\mu$g/mg | % | % |
| QA control | 3.0$\pm$ 0.5 | 3 $\pm$ 1 | 1.1 $\pm$ 0.1 | 47.7 $\pm$ 0.2 |
| QA high N | 300$\pm$ 60 | 170 $\pm$ 30 | 2.4 $\pm$ 0.5 | 48.1 $\pm$ 0.2 |
| PM control | 2.7 $\pm$ 0.8 | 2.0 $\pm$ 0.5 | 1.3 $\pm$ 0.2 | 56 $\pm$ 9 |
| PM high N | 190$\pm$ 43 | 80 $\pm$ 20 | 4.7 $\pm$ 0.2 | 45.9 $\pm$ 0.4 |

a. QA is *Q. agrifolia* and PM is *Pseudotsuga menziesii*.

**Table 2.** Summary of species-dependent foliar deposition results

| species | $R_m$ ($g_t$) s cm$^{-1}$ | $R_m$ ($g_s$)$^a$ s cm$^{-1}$ | max$^d$ V$_d$ cm s$^{-1}$ | max$^e$ $g_t^w$ mmol m$^{-2}$s$^{-1}$ | median dark V$_d$ cm s$^{-1}$ | slope$^f$ | r $g_t$ vs. slope$^g$ | [NO$_2$]$_{comp}$ ppb |
|---|---|---|---|---|---|---|---|---|
| *P. sabiniana* | 0.43±0.06$^h$ | 0.46±0.06 | 0.51±0.04 | 500±100 | 0.087 | 0.79±0.04 | -0.58$^c$ | -0.03±0.03 |
| *P.ponderosa* | 0.7±0.1 | 0.69±0.09 | 0.26±0.01 | 230±25 | 0.038 | 0.91±0.05 | -0.43$^c$ | 0.00±0.02 |
| *P.contorta* | 0.5±0.2 | 0.5±0.2 | 0.24±0.03 | 180±30 | 0.018 | 0.99±0.03 | -0.31$^c$ | 0.00±0.01 |
| *P. menziesii* | 0.30±0.07 | 0.30±0.06 | 0.26±0.02 | 230±20 | 0.044 | 0.91±0.04 | -0.30$^c$ | 0.02±0.02$^b$ |
| *C. decurrens* | 0.4±0.1 | 0.4±0.1 | 0.21±0.03 | 160±20 | 0.009 | 0.91±0.02 | -0.01 | 0.00±0.02 |
| *S. sempervirens* | 0.9±0.1 | 0.9±0.1 | 0.27±0.04 | 330±80 | 0.009 | 0.84±0.03 | -0.56$^c$ | -0.01±0.02 |
| *A. menziesii* | 0.4±0.1 | 0.4±0.1 | 0.26±0.05 | 210±10 | 0.037 | 0.93±0.03 | -0.44$^c$ | -0.02±0.01 |
| *A. macrophyllum* | 0.5±0.1 | 0.54±0.09 | 0.47±0.08 | 400±100 | 0.017 | 0.84±0.03 | -0.42$^c$ | -0.02±0.01 |
| *Q.agrifolia* | 1.3±0.3 | 1.3±0.2 | 0.15±0.01 | 90±20 | 0.008 | 0.89±0.04 | -0.14 | 0.00±0.01 |
| *Q. douglasii* | 0.2±0.1 | 0.2±0.1 | 0.30±0.03 | 180±20 | 0.004 | 0.89±0.04 | -0.24 | -0.01±0.02 |

a. $R_m$ calculated assuming $R_b = 1$ s cm$^{-1}$.

b. Statistically significant ($\alpha = 0.01$) compensation point. Compensation point listed is at limit of detection for the instrument. All other compensation points are not statistically significant ($\alpha = 0.05$).

c. Statistically significant ($\alpha = 0.05$) correlation. Correlations not indicated are not statistically significant ($\alpha = 0.05$).

d. Maximum stomatal conductance that was observed during our experiments and the error associated with that measurement.

e. Listed maximum $g_t^w$ the maximum stomatal conductance to water vapor that was observed during our experiments and the error associated with that measurement. Units in mmol m$^{-2}$s$^{-1}$ for ease of comparison with other stomatal conductance studies.

f. Total slope of measured *vs.* predicted fluxes (Fig. 2).

g. Individual slopes of predicted *vs.* measured fluxes from each day an experiment was run.

h. Calculated including data in helium.

**Table 3.** Summary of drought stress results

| tree[a] | med $\Psi_p$ (IQR)[b] MPa | med $g_t$ (IQR) cm s$^{-1}$ | med $V_d$ (IQR) cm s$^{-1}$ | $R_m$ s cm$^{-1}$ | slope[c] | $r^d$ $g_t$ vs slope | $r^e$ $\Psi_p$ vs $g_t$ |
|---|---|---|---|---|---|---|---|
| PP control | -0.60 (0.35) | 0.23 (0.17) | 0.21 (0.13) | $0.69 \pm 0.09$ | $0.89 \pm 0.02$ | $-0.59^e$ | $0.651^e$ |
| PP drought | -1.05 (0.53) | 0.07 (0.12) | 0.06 (0.12) | $0.0 \pm 0.3$ | $1.0 \pm 0.1$ | -0.10 | |
| CD control | -0.30 (0.30) | 0.13 (0.09) | 0.12 (0.09) | $0.37 \pm 0.15$ | $0.95 \pm 0.02$ | -0.11 | $0.357^e$ |
| CD drought | -0.80 (0.45) | 0.06 (0.05) | 0.06 (0.05) | $1.17 \pm 0.38$ | $0.88 \pm 0.03$ | -0.23 | |

a. PP is *Pinus ponderosa* and CD is *Calocedrus decurrens*

b. IQR is the interquartile range.

c. Slope of measured vs. predicted fluxes.

d. Pearson correlation coefficients.

e. Statistically significant ($\alpha = 0.05$ correlation).

**Table 4.** Comparison of total conductance measurements with previous works

| tree[a] | max $g_t^w$ (this study) mmol m$^{-2}$ s$^{-1}$ | reported max $g_s^w$ mmol m$^{-2}$ s$^{-1}$ | reference[a] |
|---|---|---|---|
| *P.ponderosa* | $230 \pm 25$ | 124 | Maire et al., (2015) |
| *P.contorta* | $180 \pm 30$ | $230 \pm 30$ | Arango-Velez et al. (2016)[b] |
| *P. menziesii* | $230 \pm 20$ | $140\pm 10$; 250 | Manter et al. (2000); Manter and Kavanagh (2003)[c] |
| *C. decurrens* | $160 \pm 20$ | 150 | Grantz et al. (2019)[b] |
| *S. sempervirens* | $330 \pm 80$ | 91; 240 | Maire et al., (2015); Ambrose et al., (2010) |
| *A. menziesii* | $210\pm 10$ | 150; 550 | Maire et al., (2015); Murray et al., (2019) |
| *A. macrophyllum* | $400 \pm 100$ | 420 | Murray et al., (2019) |
| *Q.agrifolia* | $90 \pm 20$ | 95; 390 | Henry et al. (2019); Murray et al., (2019) |
| *Q. douglasii* | $180 \pm 20$ | $325 \pm 30$ | Matzner et al. (2003) |

a. References respectively refer to values in the reported max $g_s^w$ column.

b. Study did not report value as a maximum stomatal conductance. The conductances shown are the maximum of the stomatal conductances reported in the cited study.

c. Theoretical calculation.