# Peer review of "Laboratory measurements of stomatal $NO_2$ deposition to native California trees and the role of forests in the $NO_x$ cycle"

_Atmospheric Chemistry and Physics, 2020_

## Referee Comment (RC1) · Anonymous Referee #1 · 16 Apr 2020

Delaria et al. report on a series of laboratory experiments to investigate whether chemical processing within the leaf limiting stomatal uptake of NO2 impacts the total NO2 exchange. Their laboratory experiments include testing the effects of nitrogen fertilization and drought. (I think the authors do these experiments to learn more about the in-leaf chemical processing, but it's not actually very clear.) They also do some multilayer canopy modeling and backhand calculations to test the large-scale impacts of their laboratory results. In general, I find the authors to take too many liberties in moving between spatial scales and in discussing the implications of their work, and the paper to be confusing and disjointed. The discussion about what happens at night is interesting but feels tangential. I think the authors need to more clearly articulate their

experimental design and how it is designed to address current information gaps or conflicting studies, as well as better connect their conclusions with their results and discuss associated uncertainties. While the paper has potential to be an important contribution to the peer reviewed literature, I can't recommend publication at this stage.

General comments. First, "deposition velocities" measured in the lab are not the same as deposition velocities from a large scale model or estimated from an eddy covariance measurement, which represent the integrated uptake below a certain height, taking into account turbulent transport. I would prefer if the authors chose another term to represent leaf-level uptake, but more importantly, this has implications for the authors' large scale modeling and backhand calculations — is it really appropriate to represent true deposition velocities with leaf-level uptake values? What about transport, leaf area, etc.?

The authors make a series of assumptions about resistances to the leaf boundary layer and cuticles in their interpretation of their laboratory results that I think need to be discussed more.

Are the authors maintaining constant temperature, pressure and humidity in the chamber over their forty minute long experiments? How might temporal variations in these quantities, or spatial variations within the chamber, affect measurements?

The canopy scale modeling and discussion in Section 4.1 is confusing. The authors do a fair amount of work in the lab to estimate Rm, and then say an increase from 0.1 s/cm to 0.6 s/cm in Rm doesn't matter based on canopy scale modeling. The paper could have just been "Rm could be off by an order of magnitude — does this matter? Let's see with a model" I guess I'm asking the authors to more clearly articulate how their setup was designed to build on present knowledge. For example, is the increase much less than they expected based on previous work?

Are there no boundary layer height products for California? I'd like to see at least some discussion of uncertainty in using only one PBL height for all of California for day or

night.

The authors use "significant" to refer to statistical testing and to emphasize the implication of a finding. This is confusing and I ask that they choose another word for the latter.

In some paragraphs multiple verb tenses are used. This is confusing.

Line comments. Line 2 - is it really absorption? Line 11-12 - what do the authors mean by effective? Line 17 - references are needed for this sentence, and the authors should specify what importance is with respect to Line 19 - "after" diffusion rather than "via" Line 28 - are the processes really happening in the mesophyll? Line 35 - a paper from 2000 isn't exactly recent Line 43-44 - "atmospherically relevant conditions" of what? Line 50 - define compensation point briefly here Line 135 - I think there needs to be a short description of Rb estimation here Line 215/219 - Rb changes with leaf morphology, leaf movement and micrometeorology. I understand Rb is hard to estimate, but I think the authors need to discuss how uncertainty in Rb may play into their results more. For example, how might inferences about stomatal and mesophyll controls be impacted by Rb variations (the authors assume constant Rb)? Line 205: is the only evidence for "believing" this measurement is consistent with a zero compensation point that the concentration is below the limit of quantification? If so, will the authors make this more clear? Line 206: I would be more careful in saying deposition of NO2 — perhaps stomatal uptake of NO2 here T deposition requires considering Rb,Ra, cuticular deposition Line 207-209: might this be affected by a lack of a diurnal cycle in light in the lab? I know there is evidence for stomatal activity at night generally, but maybe there should be some discussion of uncertainty in moving between the lab and the real world Line 210: It would be helpful if the authors explained what exactly to look for in Table 2 Line 211: the two methods don't seem that different to me — they are relying on the same assumptions — seems just like two ways of presenting one method. Line 213: and assuming zero cuticular uptake? Line 230: First, "No significant cuticular resistances" implies cuticular uptake is happening. Second, how do the authors know

СЗ

that there is no cuticular deposition when the authors are also inferring Rm? How can the authors know that the residual is Rm and not Rc? Also, I think the authors should spell out here what exactly they are suggesting that the Vd/gt ratio means ("attribute to" is a bit vague) and the assumptions involved Line 234 - spelling error Line 242-3: What do the authors mean "behave consistently"? Line 255: It would be helpful if the authors described what is observed as changing in the relationship between gt and vd, instead of just saying that there are changes and referring to a supplemental figure Line 263-6: I'm confused. My interpretation is that there is one slope for every plot in Figure 2. So how are the authors looking at a correlation between gt and the slope for each plot? The description of what the authors are doing on Lines 219-221 could be improved ("slopes were calculated from ... slopes..."). Line 284-6: Not sure what to do with this information. Line 299-300: This seems like a rather broad conclusion based on the limited evidence that the authors have presented. Line 305-6: why is the fertilized group experiencing stress "supported by previous studies [finding] a negligible impact of N fertilization on NO2 uptake"? I think "these" should refer to the sentence before "We did observe..." but the writing is unclear. Line 308: uptake can't ever be bidirectional Line 309: how do the authors know that there is actually accumulation in NO3 and NO2 within the mesophyll after fertilization? Is this from the leaf N measurements? Line 309: "neither ... nor" (here and elsewhere) Line 310: what does "disproportionation" mean? Line 311: I'm not following why this "further supports...atmospheric unimportant" Line 330: I have no idea what the authors mean "atmospherically relevant". What is/where is this discussed above? Line 340: The authors can't move like this between lab and model "deposition velocities" Line 345: not true — see 10.1002/2016JD025519 Line 339: instead of saying the models assume this, it would be more appropriate to say the Wesely scheme assumes this. Line 346: Is the box model validated for nighttime chemistry and transport in forests? Line 350: What do the authors mean at such a low degree of stomatal opening? What does "statistically equivalent" mean? That they are similar in magnitude? Line 354: Is this a range in the NOx lifetime to deposition? Or the total lifetime? Also, it doesn't seem like the authors show anything about lifetime in

Figure 6. Line 358: reference needed for major chemical nighttime sink as PAN Line 360-380: this is a lot of info to take in; please consider a table or a figure. Line 382: what are the significant inconsistencies? Line 390: seems like the authors need to say in June somewhere in the text (it's only in the figure caption). Also, why June? What years are the authors looking at for LAI and NO2? Line 397: Why do the authors use maximum vd here? It seems like the implications of this need to be emphasized. Line 398: How does one multiply by "land cover"? What are the units of "land cover"? Line 395: How big are the Forest Service plots? Do the authors define forests with less than 50% of the trees measured in the study as "nonforested"? Are they included in white space on the figure? Line 396: clarify what the effective vd is Line 398: can one get midnight measurements of NO2 from OMI? Line 400: what is chaparral? Line 406: what is significant? Line 417: when do the authors look at vapor pressure deficit? Line 419: what does "from an atmospheric perspective" mean? Line 420: I wouldn't encourage others to overlook the role of transport through turbulence and molecular diffusion at the large scale though Line 424: spelling error Line 421-5: does this really merit discussion in the very short conclusion? The authors look at different species because they have different stomatal conductances. For example, the authors say: "To test this, we measured ... over a range of stomatal conductances" in the introduction. In other words, I feel like this was the motivation in setting up the study, not a conclusion of it. Line 436: can the authors briefly summarize here their evidence for "large and important"

Figures should be cleaned up to make them more appropriate for publication. The axis labels and tick marks should look better. Figure 2 - what data is included here? No N or drought perturbations right? Figure 3 - specify acronyms used in caption; if the authors briefly described here what we are supposed to take away from helium/zero air differences that would be helpful Figure 4 - if the authors said the meaning of Vd/gt ratio in their last sentence it would be even more helpful. Generally I'm not exactly sure how to interpret this figure — what should I be looking at in terms of NH4 and NO3? Figure 5 - spelling error; again helpful to say in plain language what a compensation

**point is**

Tables could use more context/description in general Table 2 - What does Rm (gt) vs. Rm (gs) mean? Are all compensation points statistically significant or just this one? There are two "e" in the footnotes. Table 3 - Define acronym for IQR

---

## Referee Comment (RC2) · Anonymous Referee #2 · 10 Sep 2020

The mansucript of Delaria et al. investigates the potential effect of NO2 stomatal deposition to several native California tree species by using the branch enclosure techniques. They measure NO2 fluxes, deposition velocities and stomatal conductance by adding water vapor and NO2 to a gas stream passing through the branch enclosure. Additionally the effects on mesophyllic processes and foliar deposition of NO2 from excess soil nitrogen and drought stress are determined. The authors also provide some basic modeling approach to investigate the potential impact on the NOx budget in the region of California.

The methods and the results are presented more or less clearly in the manuscript

however some important aspects are still missing and are addressed in the comments below. I also have the impression that too many aspects are tried to discuss in the paper and that overall the paper would benefit in a more clear structure to guide the reader through the different aspects. Overall the results and implications are potentially important to understand the effect trees can have on the NOx burden in the atmosphere and determining if trees are sink and/or sources for NOx. Therefore the manuscript fits the scope of ACP and I recommend publications after the following comments are addressed.

General comments

Since the accurate determination of the flux of NO2 and the deposition velocity depends on the measurement of the concentration of the ingoing and outgoing air of the branch enclosure I miss a more detailed assessment how leak tight the chamber actually was. It is only stated that the chamber was operated at a slight over pressure to ensure lab air contamination. However what about leaks through which NO2 could escape? Additionally if you have higher relative humidity, how much water might condense on the Teflon wall? Might the potential water deposition on the walls depended of the mole fraction of water vapor in the chamber? What really would be beneficial to add measurements of an empty branch enclosure and measuring if and potentially how much NO2 and water vapor are lost due to leakage and/or wall losses.

Specific comments

Line 221ff and figure 2: "Some experiments were excluded (shown in red in Fig. 2), as they were determined to be outliers by a generalized extreme studentized deviate test for outliers." I am confused on how this approach was really applied to the data. While the data for P. contorta, P. menziesii, A. menziesii, A. macrophyllum, Q. agrifolia, and Q. douglasii show outliers which seem to have strangely also a linear correlation in themselves, no outliers could be found for C. decurrens and S. sempervirens. If the outliers would a result of what the authors state "most likely due to systematic error in

calibration of the Licor-7000 instrument" then I would expect the outliers to be more random and found for all data sets since I guess the Licor data was taken on the same days for all plants with one calibration applied. The finding and excluding of the outliers (which would have quite an impact if taken into account for the fitting of the measured vs. predicted fluxes (e.g. strongly for P.contorte)) needs to be discussed in more detail as to why the outliers are not more randomly distributed and seem to have a correlation in themselves.

Line 264: you examine the correlation of the total conductance vs. the slope of measured vs predicted fluxes. Why do you not provide the correlation graphs (e.g. in the supplement) as well? Seeing the correlation graphs with the fits derived from it are more instructive than just giving the numbers.

Line 268: "All tree species except for C. decurrens, Q. agrifolia, and Q. douglasii show statistically significant correlations ($\alpha$ = 0.05) (Table 2)." I have difficulties to reconcile this with Table 2. The footnote "c" indicates statistically relevant correlations however the marked values do not correspond with the tree species mentioned in the text. To restate my previous comment also to estimate this the reader would very much benefit from being able to see the correlation plots for g_t vs. slope themselves.

Line 410: In the discussion only the comparable lifetime is mentioned. However comparing Fig. 7 and Fig. 8 one also sees that the flux predicted by the model is significantly lower than during the day. So the total loss even with similar lifetime during the day will not be as much as during day time. That should be also mentioned in the discussion as well and in general the modelling of the night time fluxes and NO2 lifetime is so shortly presented and discussed that it almost appear as if an addendum. The discussion should be extended.

Line 425: "large and important" form the comments mentioned before I don't see that yet this statement can be made without at least summing up what this is based on here again.

Technical comments

Line 27: The sentence "Although the role..." is very hard to follow. I would suggest splitting the sentence in two shorter ones.

Line 159: I assume that in the sentence "100, 200, 100, and 500 $\mu$L of 0.2 M citrate, 5 mM nitroprusside,..." the second "100" is actually meant to be either 300 or 400? Otherwise is it not clear to me why the 100 is repeated.

Line 409: "The lifetimes to deposition during the day..." should read "night"

Table 2: the footnotes have two times the indicator "e". The first "e" should maybe be a "d" but the description would also not fit to the "max Vd" in the table. Please correct.

---

## Author Comment (AC1) · 24 Sep 2020

**Response to reviewer #1**

We thank the reviewer for comments and regret that he/she found our approach unclear. We have tried to clarify our thinking throughout the manuscript. In contrast to comments of the reviewer, we think the discussion of leaf-level results in the context of ecosystem scale is essential to placing the results in the context of the current understanding. We have addressed the stated concerns raised by reviewer #1 below. **Bold text** identifies the reviewer comments and our responses are in standard text.

[Figure]

Line numbers in our responses refer to the revised manuscript.

**General comments. First,"deposition velocities" measured in the lab are not the same as deposition velocities from a large scale model or estimated from an eddy covariance measurement, which represent the integrated uptake below a certain height, taking into account turbulent transport. I would prefer if the authors chose another term to represent leaf-level uptake, but more importantly, this has implications for the authors' large scale modeling and backhand calculations is it really appropriate to represent true deposition velocities with leaf-level uptake values? What about transport, leaf area, etc.?**

The term "deposition velocities" is widely used in the leaf-level literature (e.g. Teklemariam and Sparks, 2006; Chaparro-Suarez et al., 2011; Breuninger et al., 2012 etc.). Canopy fluxes are calculated as $F = V_d \times LAI \times [NO_2]$, so these canopy-level deposition velocities represent average leaf deposition velocities, as in the Big-Leaf model. We agree that, of course, vertical transport, attenuation of above-canopy light, etc. complicates canopy-level $V_d$. However, our previously published (Delaria and Cohen, 2020) canopy scale model does take into account all of these effects. This previously published and peer-reviewed model was constructed for the purpose of scaling up leaf-level processes to the canopy scale, as is discussed extensively in Delaria and Cohen, 2020. Leaf-level processes will indeed affect canopy-scale processes. Our "backhand estimations" made in section 4.5, are intended to provide the reader with a qualitative suggestion of areas that may be influenced by large deposition fluxes of $NO_2$. As more sophisticated models have shown that leaf-level deposition is a dominant control, we believe this is a useful qualitative representation. We do not think any reader would mistake our estimate for a full quantitative model. Nevertheless, we add the following qualifier at line 475 on page 15:

"The estimations provided here are intended only to suggest qualitative indications of where $NO_2$ deposition may be important. Because we are ignoring effects of vertical transport and light attenuation through the canopy, and because we are using maximum measured deposition velocities, the deposition reported here is likely to be an upper-bound estimate. We recommend areas where this estimated deposition is highest as regions that should be the subject of future field and large-scale modelling studies. "

**The authors make a series of assumptions about resistances to the leaf boundary layer and cuticles in their interpretation of their laboratory results that I think need to be discussed more.**

We provide a complete disscussion of the methods used to determine the boundary layer conductance in Delaria et al., 2018, which is referred to in line 150 on page 5. We further discuss the boundary layer in lines 228—232 of page 8, in which we use previous laboratory leaf-level studies to argue that our measured $R_b$ is an upper bound of the chamber $R_b$ when a branch is present. We further discuss the error ($\sim$ 6%) that would be introduced by assuming negligible boundary resistance. On line 256 of page 9, we have added a line explaining our determination of negligible cuticular resistances:

"The deposition observed with the chamber lights turned off could be explained completely by the measured stomatal conductance. Fits of the resistance model (Eq. 10) typically resulted in cuticular resistances larger than 1000 s cm$^{-1}$, and represented cuticular deposition not significantly above zero."

**Are the authors maintaining constant temperature, pressure and humidity in the chamber over their forty minute long experiments? How might temporal**

variations in these quantities, or spatial variations within the chamber, affect
measurements?

We have added a sentence to line 131 of page 5 to clarify our temperature and
humidity assumptions: " Over the course of a day the temperature and humidiy varied
by a maximum of 2 °C and 5%, respectively. These deviations were not found to be
significantly correlated with stomatal opening." We discuss how spatial variations in
temperature throughout the chamber would affect our calculations in lines 139—142
page 5.

**The canopy scale modeling and discussion in Section 4.1 is confusing. The
authors do a fair amount of work in the lab to estimate Rm, and then say an
increase from 0.1 s/cm to 0.6 s/cm in Rm doesn't matter based on canopy scale
modeling. The paper could have just been "Rm could be off by an order of
magnitude does this matter? Let's see with a model" I guess I'm asking the
authors to more clearly articulate how their setup was designed to build on
present knowledge. For example, is the increase much less than they expected
based on previous work?**

Our paper reports laboratory observations and their interpretations. These serve a
number of purposes. Among these is our effort to understand Rm. The discrepancies
existing in the literature on the role of the mesophyll are discussed in the introduc-
tion and to our thinking are important to assess. Even though Rm is shown to be
unimportant to canopy scale fluxes, it is important to thinking about the fate of NO2
once it enters pore fluids in the leaf and to reconciling previous studies that report
emission of NOx from leaves at low ambient NOx. Further, to our knowledge this is
the first study assess whether the particular number for Rm included in most chemical
tranport models is reasonable. A paragraph was added beginning line 310 on page 10

to further argue for the importance of our study:

"Our laboratory measurements of mesophyllic resistance address the uncertainty in the literature on whether reactions in the mesophyll may be consequential for $NO_2$ deposition velocities. To our knowledge, no previous study has explicitly calculated the mesophyllic resistance. Differences between leaf-level deposition velocities and stomatal conductances measured by Breuninger et al., 2013, and observations by Teklemmariam and Sparks, 2006, of the affects of leaf ascorbate on uptake rates have indicated mesophyllic reactions may be important. Additional studies (Gut et al., 2002; Eller et al., 2006; and Chaparro-Suarez et al., 2011) have also shown some evidence that between 20% and 40% of $NO_2$ deposition is under mesophyllic control. Our findings, however, suggest nearly 90% of uptake is controlled by the stomata."

**Are there no boundary layer height products for California? I'd like to see at least some discussion of uncertainty in using only one PBL height for all of California for day or night**

As we have stated previously, these calculations are meant to give a qualitative look at areas where deposition of $NO_2$ may be particularly important. Even so, we have adapted out figure to use a WRF-Chem output of boundary layer heights throughout the state. This updated figure does not change our conclusions.

**The authors use "significant" to refer to statistical testing and to emphasize the implication of a finding. This is confusing and I ask that they choose another word for the latter. In some paragraphs multiple verb tenses are used. This is confusing.**

[Figure]

We have gone through the manuscript and ensured that every instance that the word "significant" is used, we mean statistical significance. A different word is chosen every time we are trying to emphasize the implication of a finding. We have also adjusted verb tense where appropriate.

**Line 2: is it really absorption?**

The word was changed to "uptake".

**Line 11–12: what do the authors mean by effective?**

This word was removed. The choice of "effective" was used because, as we discussed elsewhere in the manuscript, there is some strong evidence in the literature of emission of NO. Because this emission is over an order of magnitude slower than $NO_2$ uptake, at atmospherically relevant conditions the net exchange of the chemical family $NO_x$ will be uni-directional.

**Line 17: references are needed for this sentence, and the authors should specify what importance is with respect to**

References have been added. We have changed the sentence to read: "The latter source is of particular importance in remote forested, and agricultural regions, where emission from soils is the primary source of $NO_x$."

**Line 19: "after" diffusion rather than "via"**

[Figure]

The change has been made.

**Line 28: are the processes really happening in the mesophyll?**

Our understanding is that mesophyllic processes occur in the mesophyll. We have changed the sentence to read "mesophyllic processes."

**Line 35: a paper from 2000 isn't exactly recent**

This citation has been removed from the sentence.

**Line 43–44: "atmospherically relevant conditions" of what?**

We mean under atmospherically relevant temperature, relative humidity, soil N levels, soil $NO_x$ levels, pressure, and that no modifications were made to the plants. We feel that it would not be helpful to the reader to list all conditions that were maintained at atmospheric relevance for all above studies. We have, however, removed this phrase to avoid any further confusion.

**Line 50: define compensation point briefly here**

The phrase here has been changed to "$NO_2$ emissions".

**Line 135: I think there needs to be a short description of Rb estimation here**

We have moved a sentence from section 3.1. The sentences now read:

"The boundary layer resistance to water vapor was estimated to be negligible under our experimental conditions, with an upper bound of $0.6$ s cm$^{-1}$. This was calculated by measuring the deposition of NO$_2$ to a 30 cm$^2$ tray of activated charcoal and confirmed by measuring the evaporation from a water-soaked Whatman No. 1 filter paper (Delaria et al., 2018). A detailed description of our assumption of negligible $R_b$ can be found in section 3.1."

**Line 215/219: Rb changes with leaf morphology, leaf movement and microme-teorology. I understand Rb is hard to estimate, but I think the authors need to discuss how uncertainty in Rb may play into their results more. For example, how might inferences about stomatal and mesophyll controls be impacted by Rb variations (the authors assume constant Rb)?**

We have included a more extensive discussion of $R_b$. The paragraph now reads:

"We utilized two methods for analysing the importance of the mesophyllic resistance to the deposition of NO$_2$. Figure 2 shows the predicted stomatal-limited NO$_2$ deposition fluxes, assuming negligible $R_b$ and $R_m$ ($Flux = g_t[NO_2]_{out}$) plotted vs. the measured NO$_2$ fluxes. Our upper bound measurement of $R_b$ for NO$_2$ was 1 s cm$^{-1}$ (0.6 s cm$^{-1}$ for water vapor). Assuming $g_s = g_t$ would lead to a maximum of a 60% or 10% error in the calculated $g_s$ with a $g_t$ = 0.6 cm s$^{-1}$ or $g_t$ = 0.1 cm s$^{-1}$, respectively. However, $R_b$ decreases with the enclosed leaf area according to Pape et al., 2009, which at a minimum was 200 cm$^2$. The maximum $R_b$ in the chamber should have thus been $\approx$0.1 s cm$^{-1}$. Assuming $g_s = g_t$ would lead to a maximum of a 6% error at $g_t$ = 0.6 cm s$^{-1}$ in this case. Any deviation from unity in the observed slope of predicted vs. measured

fluxes can thus be attributed to $R_m$. Any error in our assumption of negligible $R_b$ may partially mask the affect of $R_m$. We do not expect that variation in $R_b$ due to changes in leaf morphology, micrometeorology, and leaf movement would substantially change the affect of $R_b$, although we cannot rule out the possibility that this was partially responsible for day-to-day fluctuations in $NO_2$ fluxes. We confirmed the validity of our assumption of negligible $R_b$ by comparing measurements of total conductance, $g_t$, in the chamber to measurements of stomatal conductance for the enclosed branch with a Licor-6800 instrument under identical environmental conditions of light irradiation, humidity, and temperature. This test was performed on one individual of three different tree species, and in all cases the chamber $g_t$ measurements were found to be approximately equal to the Licor-6800 measurements of $g_s$ within the range of uncertainty in $g_t$. "

**Line 205: is the only evidence for "believing" this measurement is consistent with a zero compensation point that the concentration is below the limit of quantification? If so, will the authors make this more clear?**

We believe our logic on this point is fully explained. We have slightly altered the phrasing of this sentence.

**Line 206: I would be more careful in saying deposition of NO2 perhaps stomatal uptake of NO2 here deposition requires considering Rb,Ra, cuticular deposition**

We do consider all of these in our chamber, which are stated and explained.

**Line 207–209: might this be affected by a lack of a diurnal cycle in light in the lab? I know there is evidence for stomatal activity at night generally, but maybe**

**there should be some discussion of uncertainty in moving between the lab and the real world.**

There is a diurnal cycle of lights on and lights off on a 12 h light/dark period (section 2.1). Our results are also consistent with previous experiments in the field of leaf-level stomatal closure at night. We do observe slow closing and opening of the stomata when the lights are turned on or off, such that it takes approximately 1—2 hours for the stomata to reach minimum or maximum opening. We only considered data after the stomatal response had stabilized. We are not aware of any physiological evidence that there would be any differences between the lab and the real world due to sudden changes in light rather than gradual setting and rising of the sun, except during this transition time.

**Line 210: It would be helpful if the authors explained what exactly to look for in Table 2**

All results discussed are in table 2. Specifics of what to look for in table 2 are discussed thoughout the manuscript.

**Line 211: the two methods don't seem that different to me they are relying on the same assumptions seems just like two ways of presenting one method.**

The first discussed shows the overall deposition velocity stomatal scaling factor determined from all data points from all experiments. This method allows the reader to see the overall importance of the mesophyll. The second visualization method allows for a more explicit calculation of mesophyllic resistance. We believe both methods are helpful for communicating our conclusions even thought they are similar.

**Line 213: and assuming zero cuticular uptake?**

Yes, this has been added.

**Line 230: First, "No significant cuticular resistances" implies cuticular uptake is happening. Second, how do the authors know that there is no cuticular deposition when the authors are also inferring Rm? How can the authors know that the residual is Rm and not Rc? Also, I think the authors should spell out here what exactly they are suggesting that the Vd/gt ratio means ("attribute to" is a bit vague) and the assumptions involved**

We have changed the wording to be: "No evidence of cuticular deposition was observed".

The description of $V_d/g_t$ ratio has been changed for clarity. It now reads:

"Positive y-intercepts are indications of cuticular deposition and curvatures in the fit away from the 1:1 line are implications of mesophyllic resistance. "

**Line 234: spelling error**

This has been corrected.

**Line 242-3: What do the authors mean "behave consistently"?**

This sentence has been removed.

**Line 255: It would be helpful if the authors described what is observed as changing in the relationship between gt and vd, instead of just saying that there are changes and referring to a supplemental figure**

This sentence has been deleted to avoid further confusion and a reference to the figure is included in the previous sentence. This figure is similar to Figure 3 and was used to calculate $R_m$.

**Line 263-6: I'm confused. My interpretation is that there is one slope for every plot in Figure 2. So how are the authors looking at a correlation between gt and the slope for each plot? The description of what the authors are doing on n Lines 219-221 could be improved ("slopes were calculated from . . . slopes. . .").**

There is one slope for every plot, which often contains over 20 days of experiments. This slope is calculated as a weighted average of the slopes from each day of experiments.

Lines 219-221 now read (now beginning line 242 in the revised manuscript) :

"Figure 2 shows each flux measurement as a single data point. For each day of experiments a slope of predicted vs. measured fluxes was obtained from a least squares cubic weighted fit for the 8—12 fluxes measured at varying $NO_2$ concentrations. The reported slope for a given species (shown in blue in Fig. 2) was

calculated using a weighted average of the slopes from all experiment days. This was done to minimize the contribution of systematic errors potentially introduced by the Licor 7000 instrument, which was calibrated daily. All data points for a given day were excluded (shown in red in Fig. 2) if the calculated slope on that day was determined to be an outlier by a generalized extreme studentized deviate test for outliers."

Lines 263-6 now read (beginning line 290 in revised manuscript):

"We also examined the potential impact of the mesophyllic processing of $NO_2$ by considering the Pearson's correlation coefficient between $g_t$ and the slope for an individual experiment (1 day of light or dark data) of measured vs. predicted fluxes."

**Line 284-6: Not sure what to do with this information.**

We include this to compare our results to what atmospheric models currently include. We discuss the implications in the subsequent text.

**Line 299-300: This seems like a rather broad conclusion based on the limited evidence that the authors have presented.**

Our use of the word "suggest" rather than a stronger one is intended to encourage the reader to make their own judgement. We think the statement appropriate based on the evidence and analysis we present.

Nevertheless we have clarified the sentence to make our conclusions more specific to California forests (line 332 -335 in the revised manuscript):

"Contributions from mesophyllic processing, though mechanistically important at a cellular level, are likely to not matter at the canopy-scale in California forests. We therefore suggest that on canopy and regional scales, mesophyllic processes within leaves of trees represent a negligible contribution to $NO_x$ budgets and lifetimes in California. More studies on crops, grasses, and North American tree species from outside of California are needed."

**. Line 305-6: why is the fertilized group experiencing stress "supported by previous studies [finding] a negligible impact of N fertilization on NO2 uptake"? I think "these" should refer to the sentence before "We did observe. . ." but the writing is unclear.**

Sentences have been rearranged for clarity:

"We observed no effects of soil nitrogen, in the form of $NH_4^+$ and $NO_3^-$, or the leaf nitrogen content on the ratio of $V_d/g_t$ (Fig. 4) for either *Q. agrifolia* or *P. menziesii.* Changes in this ratio would indicate an effect on the mesophyllic resistance. We did observe declines in $g_t$ in the fertilized group relative to the control group during the later stages of experimentation, which coincided with observable evidence of plant stress (e.g., browning, wilting, and beginning signs of embolism). All variation in the uptake rates ($V_d$) could be explained exclusively with deviations in $g_t$. These results are supported by previous studies which have also found a negligible impact of nitrogen fertilization on $NO_2$ uptake (Teklemmariam and Sparks 2006; Joensuu et al., 2014). "

**Line 308: uptake can't ever be bidirectional**

"bidirectional" has been changed to "reversible".

**Line 309: how do the authors know that there is actually accumulation in NO3 and NO2 within the mesophyll after fertilization? Is this from the leaf N measurements?**

Based on the leaf N measurements we can say that either we accumulated inorganic nitrogen in the leaves and it had no effect, or that we gave the an extreme amount of nitrogen fertilizer and it still did not cause accumulation. The sentence (line 343 in revised manuscript) has been adjusted to make this more clear:

"If the fertilizer results in increased $NO_3^-$ and $NO_2^-$ in the leaf, this suggests that the mechanism of $NO_2$ uptake via dissolution and subsequent reduction of $NO_3^-$ and $NO_2^-$ is likely not reversible and not influenced by accumulation of $NO_3^-$ and $NO_2^-$ within the mesophyll. Alternatively, if the increase in soil nitrogen leads only to an accumulation of organic nitrogen in the leaf, this increase has no effect on the uptake rates."

**? Line 309: "neither . . . nor" (here and elsewhere)**

Fixed.

**Line 310: what does "disproportionation" mean?**

Disproportionation is the chemical word for a reaction of the form $2A \rightarrow A' + A''$, where substance A is simultaneously oxidized and reduced (See Lee and Schwarz 1981). Here $2NO_2 \rightarrow$ nitrate and nitrite.

[Figure]

**Line 311: I'm not following why this "further supports. . .atmospheric unimportant"**

The following has been added to replace the sentence previously on line 311 (347 in revised manuscript):

"Based on our current understanding of the mechanism of $NO_2$ mesophyllic processing, if reactions in the mesophyll indeed affect the rate of stomatal uptake, our fertilization experiments should have succeeded in changing $NO_2$ uptake rates, given that they succeeded in changing leaf nitrogen content. Because we observed no effect of nitrogen fertilization on $NO_2$ uptake, we believe that this finding further supports that reactions within the mesophyll may be atmospherically unimportant. It is also possible, that the disproportionation of $NO_2$ to form nitrate and nitrite, and scavenging by antioxidants (e.g. ascorbate) are the rate limiting steps in the mesophyllic processing of $NO_2$."

**Line 330: I have no idea what the authors mean "atmospherically relevant". What is/where is this discussed above?**

See sections 4.1, lines 315—325 in the revised manuscript. We revise the sentence as follows:

"Although there was a statistically significant impact of drought stress on $R_m$, this is unlikely to be important to the overall uptake rates of $NO_2$ an the canopy scale for reasons discussed in section 4.1."

**Line 340: The authors can't move like this between lab and model "deposition**

**velocities"**

We do not use the term "deposition velocities" here, or anywhere in this paragraph. The studies cited here all infer that deposition to leaves or soils are necessary to describe observed canopy fluxes and mixing ratios of $NO_x$. Leaf-level deposition does have an effect on canopy-scale processes.

**Line 345: not true see 10.1002/2016JD025519**

We thank the reviewer for pointing out this study. A citation to this reference has been added :

" Sparks et al., 2013 did not observe any evidence of non-stomatal deposition in the laboratory, but more recently Sun et al., 2016 implicated non-stomatal deposition in accounting for over 20% of PAN leaf-level deposition. Our PAN deposition experiments however, discussed in Place et al., EST in press, also did not identify any significant non-stomatal deposition. Despite the existing differences regarding the importance of non-stomatal PAN deposition, we suggest that a significant portion of the "missing" deposition sink of $NO_2$ and peroxyacyl nitrates at night may be due to non-total closure of the stomata. "

**Line 339: instead of saying the models assume this, it would be more appropriate to say Wesely scheme assumes this.**

This has been adjusted.

**Line 346: Is the box model validated for nighttime chemistry and transport in forests?**

Yes. Delaria and Cohen, 2020 compared the box model to field measurements over a 24 hour period. In developing that model we went through additional validation processes where we ensured that the resulting lifetimes and loss rates calculated with the model at all times of day were reasonable when compared with field measurements.

**Line 350: What do the authors mean at such a low degree of stomatal opening? What does "statistically equivalent" mean? That they are similar in magnitude?**

The sentence has been edited to read: "At such low stomatal conductances, we found these deposition velocities to be not significantly different ($\alpha = 0.05$) from the stomatal conductance to $NO_2$."

**Line 354: Is this a range in the NOx lifetime to deposition? Or the total lifetime? Also, it doesn't seem like the authors show anything about lifetime in Figure 6.**

This has been corrected.

**Line 358: reference needed for major chemical nighttime sink as PAN**

References have been added.

**360-380: this is a lot of info to take in; please consider a table or a figure.**

A table has been added to the revised manuscript.

**Line 382: what are the significant inconsistencies?**

The inconsistencies were outlined in the previous paragraph. There are several contrasting gmax measured by the studies referenced.

**Line 390: seems like the authors need to say in June somewhere in the text (it's only in the figure caption). Also, why June? What years are the authors looking at for LAI and NO2?**

The information has been added to the text.

**? Line 397: Why do the authors use maximum vd here? It seems like the implications of this need to be emphasized.**

We have added additional discussion at this point in the manuscript. We use maximum because our purpose is to illustrate the importance of the deposition in a consistent way across the domain. Our intention is that this "back of the envelope" calculation might be used by others to think about locations where deposition would be interesting to explore further.

**Line 398: How does one multiply by "land cover"? What are the units of "land cover"?**

This has been removed from the equation. Landcover was either nan for not forest, or

1 for forest, but this is covered by the sentence: "Only forested sites were considered".

**Line 395: How big are the Forest Service plots? Do the authors define forests with less than 50% of the trees measured in the study as "nonforested"? Are they included in white space on the figure?**

This information has been added to the manuscript. They would not be in the white space because the plots are interpolated to a 500 m grid.

**Line 396: clarify what the effective vd is**

The line has been corrected to:

"For each approximately 24 km$^2$ hexagonal plot (Bechtold et al., 2005) in the Forest Service Inventory that contained more than 50% of the trees measured in our study, an effective deposition velocity to $NO_2$ ($V_d^{eff}$) was calculated as a weighted (by tree species abundance) average from the $V_d^{max}$ values listed in Table 2 (Fig. S3)."

**Line 398: can one get midnight measurements of NO2 from OMI?**

No. Our midnight measurements were from a WRF-CHEM simulation. We have corrected this in the manuscript, and re-calculated deposition fluxes during both the day and night using the $NO_2$ and PBL outputs from this simulation for consistency.

**Line 400: what is chaparral?**

[Figure]

It is a biome found in southern California, characterized by drought-resistant broad-leaved evergreen shrubs and trees (often oaks). The climate consists of hot dry summers and mild wet winters. There is also frequent drought and fire in these regions.

**Line 406: what is significant?**

This has been clarified in the marked-up manuscript.

**Line 417: when do the authors look at vapor pressure deficit?**

We alter the stomatal conductance by changing the chamber humidity under the same temperature conditions, which necessarily means we are changing the vapor pressure deficit. Nevertheless we have changed "vapor pressure deficit" here to "relative humidity" for consistency.

**Line 419: what does "from an atmospheric perspective" mean?**

This was added to contrast from a cellular and plant physiological perspective, where there might be indeed variations of internal processing of $NO_2$.

**Line 420: I wouldn't encourage others to overlook the role of transport through turbulence and molecular diffusion at the large scale though**

We do not believe we are doing so. We have changed the sentence to:

"This opens the possibility of using direct measurements of stomatal conductance–coupled with models and measurements of chemical transport, known relationships of the effects of environmental conditions on stomatal opening, measurements of canopy conductance, as well as indirect measurements–such as satellite solar-induced fluorescence–to infer $NO_x$ foliar exchange."

**Line 424: spelling error**

This has been fixed.

**Line 421-5: does this really merit discussion in the very short conclusion? The authors look at different species because they have different stomatal conductances. For example, the authors say: "To test this, we measured . . . over a range of stomatal conductances" in the introduction. In other words, I feel like this was the motivation in setting up the study, not a conclusion of it.**

The differences in these species have not been shown before, and many of them–our six conifers, two broadleaf deciduous trees, and two broadleaf evergreen trees–would be treated the same in the widely utilized Wesely model. The range of stomatal conductance was achieved for each of the ten species by varying humidity, as is discussed in the methods sections, and demonstrated in Figure 3.

**Line 436: can the authors briefly summarize here their evidence for "large and important"**

We have changed the wording to:

"Our observations of stomatal opening in the absence of light also suggest foliar deposition may represent as much as 25% of the total $NO_x$ loss at night, with stomatal deposition velocities as high as 0.038 cm s$^{-1}$. "

**Figures should be cleaned up to make them more appropriate for publication. The axis labels and tick marks should look better.**

We will review the figures in the galleys to ensure that labels and tick marks are clear to the reader.

**Figure 2: what data is included here? No N or drought perturbations right?**

This figure does include N and drought data. The figure caption has been updated to clarify this.

**Figure 3: specify acronyms used in caption; if the authors briefly described here what we are supposed to take away from helium/zero air differences that would be helpful**

These corrections have been made in the revised figure caption.

**Figure 4: if the authors said the meaning of Vd/gt ratio in their last sentence it would be even more helpful. Generally I'm not exactly sure how to interpret this figure what should I be looking at in terms of NH4 and NO3?**

The conclusions based on this figure are discussed in the text. Ideally, the captions should not have interpretation of figures, just describe the content. Nevertheless, we add: "The amount of soil and leaf nitrogen has no significant impact on the $V_d/g_t$ ratio." and revise the caption to read:

"The $V_d/g_t$ ratio is plotted against soil nitrogen concentration in the form of $NH_4^+$ and $NO_3^-$ for (a) *Q. agrifolia* and (c) *P. menziesii*. The dashed line shows a linear fit to $NH_4^+$ data. The relationship is not significantly different ($\alpha$ = 0.05) when fit to $NO_3^-$ data. The $V_d/g_t$ ratio is plotted against the leaf nitrogen:carbon ratio for (b) *Q. agrifolia* and (d) *P. menziesii*. $V_d/g_t$ ratios less that 1 imply contributions from the mesophyll to the $NO_2$ uptake rate. On each pannel the Pearson's correlation coefficient and the p-value for the slope are shown. The amount of soil and leaf nitrogen has no significant impact on the $V_d/g_t$ ratio."

**Figure 5: spelling error; again helpful to say in plain language what a compensation point is**

The error has been corrected and a definition added.

**Table 2 - What does Rm (gt) vs. Rm (gs) mean? Are all compensation points statistically significant or just this one? There are two "e" in the footnotes.**

Only the one identified is statistically significant. Clarifications have been made in the table footnotes.

**Table 3 - Define acronym for IQR**

The acronym has been defined.

---

## Author Comment (AC2) · 24 Sep 2020

**Response to reviewer #2**

We thank reviewer 2 for their constructive comments. We have addressed the stated concerns below. **Bold text** identifies the reviewer comments and our responses are in standard text. Line numbers in our responses refer to the revised manuscript.

**General comments: Since the accurate determination of the flux of NO2 and the deposition velocity depends on the measurement of the concentration of**

**the ingoing and outgoing air of the branch enclosure I miss a more detailed assessment how leak tight the chamber actually was. It is only stated that the chamber was operated at a slight over pressure to ensure lab air contamination. However what about leaks through which NO2 could escape? Additionally if you have higher relative humidity, how much water might condense on the Teflon wall? Might the potential water deposition on the walls depended of the mole fraction of water vapor in the chamber? What really would be beneficial to add measurements of an empty branch enclosure and measuring if and potentially how much NO2 and water vapor are lost due to leakage and/or wall losses.**

We provide a more detailed description of our chamber setup in Delaria et al., 2018. With the dynamic chamber setup an equilibrium is reached where the rate of air entering and leaving the chamber is equal. Some of the air leaving the chamber is sampled in our system and some leaks out of the chamber. The leaking out of the chamber does not matter so long as there is positive pressure in the chamber to prevent laboratory air from entering the chamber. We calculate the deposition fluxes after the chamber has reached this equilibrium. We also maintain our chamber to below 90% relative humidity to minimize chamber condensation. To account for wall losses of both NO$_2$ and water vapor, we periodically ($\approx$ monthly) measure the wall loss of these compounds and use this to correct our calculations. With the lifetime in our chamber around 2 min, the wall loss of NO$_2$ is approximately 2%. We have added the following statements to the revised manuscript lines P3 83—85, P4, 92—95 and P5, 124—127, respectively. :

"where [NO$_2$]$_{in}$ and [NO$_2$]$_{out}$ are concentrations of NO$_2$ entering and exiting the chamber at chamber equilibrium, respectively. Chamber equilibrium is achieved when the flow rates in and out of the chamber are equal and can be identified by a constant concentration of [NO$_2$]$_{out}$."

"Experiments to an empty chamber were conducted approximately every two months during this study to calculate the deposition of $NO_2$ to the chamber walls. The wall loss was at maximum $\sim$2% of the $[NO_2]_{in}$ concentration and was background subtracted from our flux calculations."

"Measurements of an empty chamber were also used to calculate and correct for the water vapor deposition to the chamber at varying relative humidity. The difference between $\omega_a$ and $\omega_e$ for an empty chamber was not statistically significant and at all relative humidity levels was within instrumental uncertainty of the Licor-6262."

**Line 221 and figure 2: "Some experiments were excluded (shown in red in Fig. 2),as they were determined to be outliers by a generalized extreme studentized deviate test for outliers." I am confused on how this approach was really applied to the data. While the data for P. contorta, P. menziesii, A. menziesii, A. macrophyllum, Q. agrifolia, and Q. douglasii show outliers which seem to have strangely also a linear correlation in themselves, no outliers could be found for C. decurrens and S. sempervirens. If the would a result of what the authors state "most likely due to systematic error in calibration of the Licor-7000 instrument" then I would expect the outliers to be more random and found for all data sets since I guess the Licor data was taken on the same days for all plants with one calibration applied. The finding and excluding of the outliers (which would have quite an impact if taken into account for the fitting of the measured vs. predicted fluxes (e.g. strongly for P.contorte)) needs to be discussed in more detail as to why the outliers are not more randomly distributed and seem to have a correlation in themselves.**

Following comments also from reviewer #1, we have made adjustments to our discussion of Figure 2 to clarify the methods used.

Figure 2 shows each flux measurement we made as a single data point. During each day of experiments we made a 8—12 different flux measurements at different $NO_2$ concentrations. The licor instruments were calibrated each day and a different water vapor concentration was delivered to the chamber. A slope was individually calculated for each day. Red outlier points are all the data points for a given day having a slope determined to be an outlier. We did it this way because we occasionally noticed issues with a daily Licor calibration.

Lines 239—247 :

"Figure 2 shows each flux measurement as a single data point. For each day of experiments a slope of predicted vs. measured fluxes was obtained from a least squares cubic weighted fit for the 8—12 fluxes measured at varying $NO_2$ concentrations. The reported slope for a given species (shown in blue in Fig. 2) was calculated using a weighted average of the slopes from all experiment days. This was done to minimize the contribution of systematic errors potentially introduced by the Licor instruments, which were calibrated daily. All data points for a given day were excluded (shown in red in Fig. 2) if the calculated slope on that day was determined to be an outlier by a generalized extreme studentized deviate test for outliers."

**Line 264: you examine the correlation of the total conductance vs. the slope of measured vs predicted fluxes. Why do you not provide the correlation graphs (e.g. in the supplement) as well? Seeing the correlation graphs with the fits derived from it are more instructive than just giving the numbers.**

Figures have been added to the supplement.

**Line 268: "All tree species except for C. decurrens, Q. agrifolia, and Q. douglasii show statistically significant correlations ($\alpha = 0.05$) (Table 2)." I have difficulties to reconcile this with Table 2. The footnote "c" indicates statistically relevant correlations however the marked values do not correspond with the tree species mentioned in the text. To restate my previous comment also to estimate this the reader would very much benefit from being able to see the correlation plots for $g_t$ vs. slope themselves.**

This was an error that has been corrected in the revised manuscript. Note that the listed correlations have changed. The correlations in the table were from a previous manuscript version from before an error in our code was found. The text was correct. Our conclusions are unaffected.

**Line 410: In the discussion only the comparable lifetime is mentioned. However comparing Fig. 7 and Fig. 8 one also sees that the flux predicted by the model is significantly lower than during the day. So the total loss even with similar lifetime during the day will not be as much as during day time. That should be also mentioned in the discussion as well and in general the modelling of the night time fluxes and NO2 lifetime is so shortly presented and discussed that it almost appear as if an addendum. The discussion should be extended.**

Yes this is completely true. Our nighttime discussion is meant to suggest the deposition of NO2 is an import sink for NO2 at night that will compete with chemical loss. However, it is correct that the total flux from an ecosystem perspective would be

small. Additional discussion has been added. We have also added the following to the revised manuscript line 465—470:

"The deposition fluxes and lifetimes to deposition during the night are shown in Fig. 8. With reduced deposition velocities at night, the nighttime deposition flux and the resulting total loss of NO2to deposition is small. However, with a reduced boundary layer during the night, the lifetime of NOx to deposition is on the same order as the deposition lifetime during the day (10—100 hr) and the overall NOx lifetime at night. This indicates this loss pathway may be an important nighttime sink of NOx from the atmosphere and may affect the nighttime chemical NOx sinks of alkyl nitrate formation and N2O5 chemistry."

**Line 425: "large and important" form the comments mentioned before I don't see that yet this statement can be made without at least summing up what this is based on here again.**

This statement has been edited to read: ". Our observations of stomatal opening in the absence of light also suggest foliar deposition may represent as much as 25% of the total $NO_x$ loss at night, with stomatal deposition velocities as high as 0.038 cm s-1."

**Line 27: The sentence "Although the role. . ." is very hard to follow. I would suggest splitting the sentence in two shorter ones.**

We have replaced the sentence with: "Although the role of stomatal conductance (gs) in controlling the deposition of $NO_2$ is well-documented, the impact of mesophyllic processes remains poorly resolved. These mesophyllic mechanisms are complex and include any process taking place between the intercellular air space and the ultimate

nitrogen assimilation site."

**Line 159: I assume that in the sentence "100, 200, 100, and 500 $\mu$L of 0.2 M citrate, 5 mM nitroprusside,. . ." the second "100" is actually meant to be either 300 or 400? Otherwise is it not clear to me why the 100 is repeated.**

The numbers in the list refer to respective listed reagents. We have editted this sentence to be more clear:

"100 $\mu$L of 0.2 M citrate , 200 $\mu$L of 5 mM nitroprusside, 100 $\mu$L of 0.3 M hypochlorite reagents, and 500 $\mu$L of milli-q water were then added sequentially into each cuvette."

**Line 409: "The lifetimes to deposition during the day. . ." should read "night"**

Yes it should have read "night". This has been corrected and the section has been updated following comments from reviewer 1. Please see the marked-up manuscript.